# Mutational signature dynamics shaping the evolution of oesophageal adenocarcinoma

Sujath Abbas[1], Oriol Pich[2], Ginny Devonshire [3], Shahriar A. Zamani [1], Annalise Katz-Summercorn [1], Sarah Killcoyne[1,4], Calvin Cheah[1], Barbara Nutzinger[1], Nicola Grehan[1], Nuria Lopez-Bigas [2,5,6], OCCAMS Consortium*, Rebecca C. Fitzgerald [1,37] ✉ & Maria Secrier [7,37] ✉

A variety of mutational processes drive cancer development, but their dynamics across the entire disease spectrum from pre-cancerous to advanced neoplasia are poorly understood. We explore the mutagenic processes shaping oesophageal adenocarcinoma tumorigenesis in 997 instances comprising distinct stages of this malignancy, from Barrett Oesophagus to primary tumours and advanced metastatic disease. The mutational landscape is dominated by the C[T > C/G]T substitution enriched signatures SBS17a/b, which are linked with *TP53* mutations, increased proliferation, genomic instability and disease progression. The APOBEC mutagenesis signature is a weak but persistent signal amplified in primary tumours. We also identify prevalent alterations in DNA damage repair pathways, with homologous recombination, base and nucleotide excision repair and translesion synthesis mutated in up to 50% of the cohort, and surprisingly uncoupled from transcriptional activity. Among these, the presence of base excision repair deficiencies show remarkably poor prognosis in the cohort. In this work, we provide insights on the mutational aetiology and changes enabling the transition from pre-neoplastic to advanced oesophageal adenocarcinoma.

Oesophageal cancer is the sixth major cause of death globally, with more than 400,000 deaths registered in 2017, and it remains a public health challenge[1,2]. Oesophageal adenocarcinoma (OAC) is one of two main subtypes for this cancer, and its incidence has increased substantially in western developed nations over the last four decades. OAC generally presents late, when loco-regional spread has already occurred and therefore has a dismal 5-year survival rate of <20%[3]. A major risk factor for OAC is chronic gastro-oesophageal reflux disease (GORD)[4,5], which predisposes to cancer via the metaplastic precancerous stage called Barrett Oesophagus. This preneoplastic condition offers the opportunity to gain insights into the early triggers of this cancer, with previous studies showing surprisingly extensive mutational damage early on in the disease, including in Barrett Oesophagus samples from patients who do not progress to cancer[6,7].

Such DNA damage, arising from extrinsic and intrinsic mutational processes acting throughout an individual's lifetime, is imprinted in the genome of cancer cells in the form of distinct patterns of nucleotide substitutions or larger chromosomal rearrangements. These recurring patterns have enabled cancer researchers to understand how different risk factors can shape the genomes of cells towards a

[1]Early Cancer Institute, University of Cambridge, Cambridge, UK. [2]Institute for Research in Biomedicine (IRB Barcelona), The Barcelona Institute of Science and Technology, Barcelona, Spain. [3]Cancer Research UK Cambridge Institute, University of Cambridge, Cambridge, UK. [4]European Molecular Biology Laboratory, European Bioinformatics Institute (EMBL-EBI), Hinxton, UK. [5]Institució Catalana de Recerca i Estudis Avançats (ICREA), Barcelona, Spain. [6]Centro de Investigación Biomédica en Red en Cáncer (CIBERONC), Instituto de Salud Carlos III, Madrid, Spain. [7]UCL Genetics Institute, Department of Genetics, Evolution and Environment, University College London, London, UK. [37]These authors jointly supervised this work: Rebecca C. Fitzgerald, Maria Secrier. *A list of authors and their affiliations appears at the end of the paper. ✉e-mail: rcf29@cam.ac.uk; m.secrier@ucl.ac.uk

neoplastic phenotype[8–11]. The distribution of such acquired mutations is specific to their causal triggers, which can either manifest as endogenous impairment of cellular processes or as exogenous mutagens[12,13]. In their simplest form, these patterns present as single base substitutions in a trinucleotide context and are termed 'mutational signatures'.

Large scale efforts have comprehensively catalogued such mutational footprints across different cancer types and linked them with a variety of carcinogen exposures, e.g., smoking, UV light damage, but also with intrinsic DNA damage repair (DDR) defects or ageing-related processes[8,9,14]. Global collaborative studies such as the Mutographs project are working to elucidate the possible causes of cancer development and our laboratory has been involved in investigating the aetiology of oesophageal squamous cell carcinoma as part of this project[15]. Previously, we also showed that mutational signatures can be employed to delineate three distinct aetiology pathways in OAC, exhibiting hypermutated, DNA damage impaired and oxidative stress-linked phenotypes, respectively, and this classification could inform therapeutic opportunities[16]. A recent report from the PanCancer Analysis of Whole Genomes (PCAWG) consortium recapitulated these main patterns in OAC and highlighted intriguing subclonal decreases in the SBS17 signature, the dominant pattern of mutational damage observed in OAC[9,17]. However, these studies were limited to relatively small numbers of OAC cases (129 and 97, respectively), focused exclusively on primary disease and did not analyse clinical or demographic associations. Thus, this still leaves major questions unanswered around the biological role of mutational signatures throughout the entire course of OAC development from preneoplasia to metastases, their aetiology and dynamics during cancer progression, as well as the influence of various clinical risk factors on tumour emergence and mutation fixation into cancer genomes. Understanding the evolution of this cancer from pre-malignant lesions into fully developed tumours and tracing its metastatic spread will help guide its clinical management[18].

In this work, we aim to understand what mutational forces drive disease progression from pre-cancerous stages to advanced malignancy in OAC. To this end, we survey a cohort of 997 patients across different stages of OAC progression, from pre-malignant to advanced disease (Fig. 1, Supplementary Tables 1, 2). Based on the pattern of single base substitutions observed from whole-genome sequencing data, we infer the mutational processes that are likely to have acted during the evolution of this cancer and characterise their prevalence across cancer stages. We find certain mutagenic footprints could be indicative of disease stage, identify consistent evidence for specific DDR deficiencies and pinpoint evolutionary shifts in mutational processes that play a key role in shaping the progression of this disease.

## Results

### Mutational signatures from pre-malignant to advanced OAC

We employed whole-genome sequencing data from 161 Barrett Oesophagus samples, 777 OAC primary tumours and 59 metastatic samples to infer and compare the signatures of mutational processes that operate during the course of this disease (Fig. 1). We performed de novo reconstruction of mutational signatures jointly across samples in all cancer stages using Non-Negative Matrix Factorisation (NMF) via SigProfiler[8]. This analysis uncovered a total of 14 single base substitution signatures with evidence of activity in these genomes (Fig. 2a), updating the disease landscape characterised in our previous study[16]. The contributions of the various mutational processes to individual genomes were determined through multiple linear regression using deconstructSigs[19].

Across disease stages, we observed an increase in the tumour mutational burden from Barrett Oesophagus to primary tumours to metastases, as expected (Supplementary Fig. 1). Signatures SBS17a/b were the most prevalent, along with evidence for mutational processes linked with ageing (SBS1/5/40), oxidative stress (SBS18), APOBEC activity (SBS2), base excision repair mutagenesis (SBS30) and DDR impairment (SBS3/8) (Fig. 2a, Supplementary Fig. 2a–c). To confirm the latter, we performed indel signature inference (Supplementary Fig. 3) and sought contributions of ID6 and ID8 signatures in the same samples to strengthen evidence for homologous recombination (HR) and double strand break repair defects. At a conservative threshold of >5% for all these signatures, 61 samples were found to be HR deficient, with an enrichment of such defects in primary tumours (8% of cases, Chi-square test 11.371, df = 2, $p = 0.003$, Supplementary Table 3).

We also observed evidence of mismatch repair (MMR) deficiency (SBS44) in 35 primary tumours (4.5%) as well as one Barrett Oesophagus sample. MMR defects have been linked with hypermutated genomes, increased neoantigen presentation, immune evasion and improved responses to checkpoint inhibition in a variety of cancers[20]. We also confirmed that in our cohort the MMR deficient samples presented significantly higher mutational burden, as well as higher

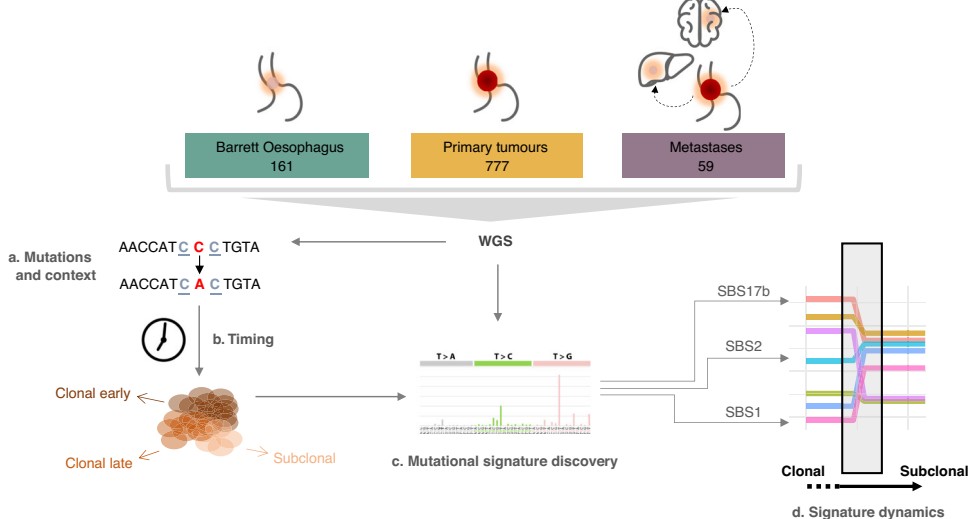

**Fig. 1 | Study workflow.** Samples from pre-cancerous, primary and metastatic stages were whole-genome sequenced (WGS) and mutations were called (**a**). Mutations are timed and categorised into clonal early/late or subclonal (**b**) before mutational signature analysis (**c**). Finally, the dynamics of mutational processes are studied in relation to the clonal composition of the samples (**d**).

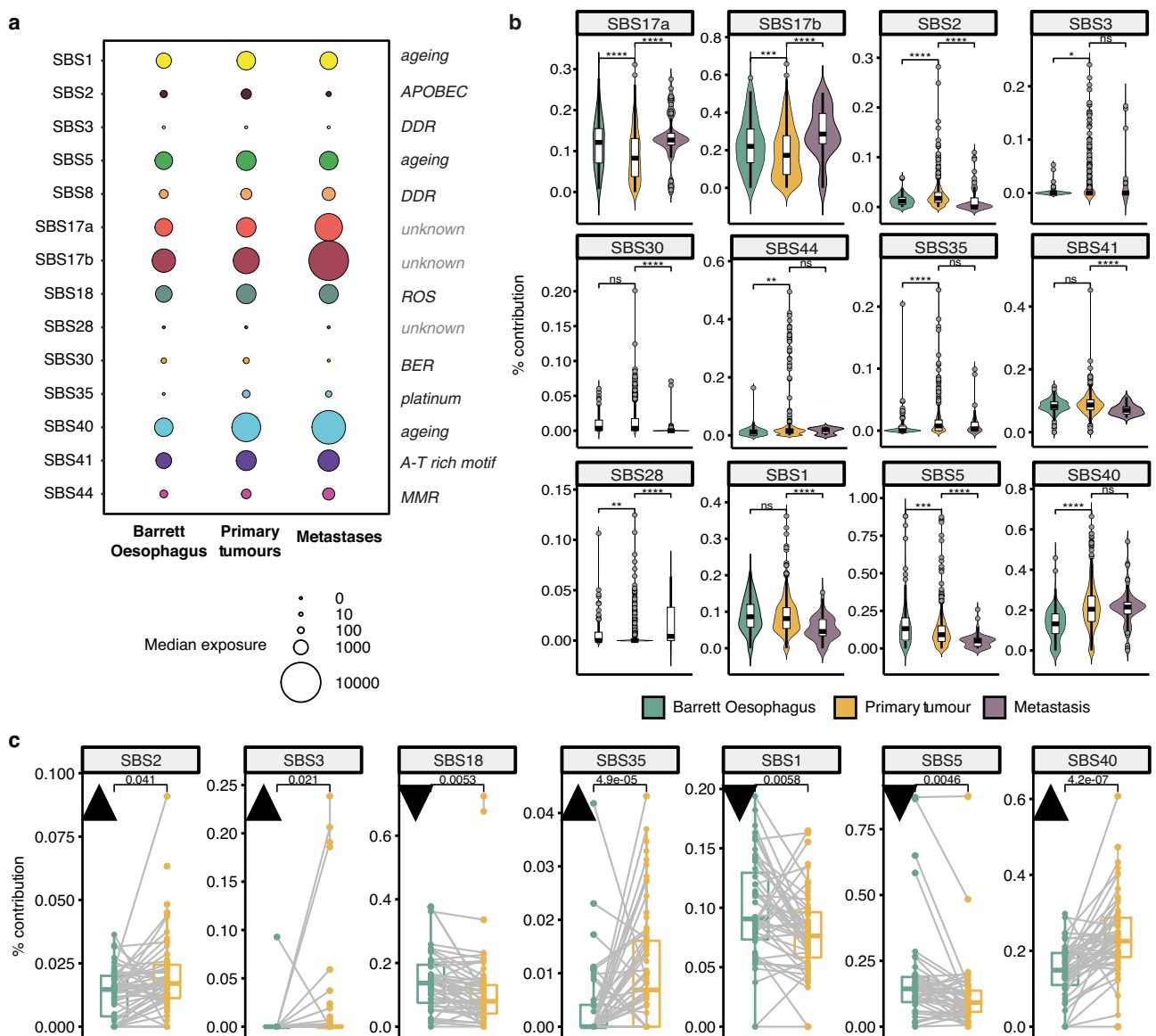

**Fig. 2 | Mutational signature landscape across disease stages. a** The median prevalence of mutational signatures present identified in the three disease stages (Barrett Oesophagus, *n* = 161; primary tumours, *n* = 777; metastases, *n* = 59). The magnitude of the circles is proportional to the median number of mutations contributed by a specific signature to samples within a disease cohort. The causative factor underlying each signature is detailed to the right of the plot when known. DDR = DNA damage repair deficiency; ROS = reactive oxygen species; BER = base excision repair; MMR = mismatch repair deficiency. **b** Mutational signature contributions compared across the three disease conditions in non-matched, independently measured samples (Barrett Oesophagus, *n* = 114, green; primary tumours, *n* = 706, yellow; metastases, *n* = 55, purple). Only signatures that show a significant change in at least one disease stage are shown (signatures 8 and 18 were stable across disease stages). The centerline of boxes depicts the median values; the bottom and top box edges correspond to the first and third quartiles. The upper

and lower whiskers extend from the hinges to the largest and smallest values, respectively, no further than 1.5* the inter-quartile range. Two-sided Wilcoxon signed-rank test *p*-values are displayed (not adjusted for multiple comparisons). Exact *p*-values are provided in the Source Data file. **c** Changes in mutational signature prevalence between matched Barrett Oesophagus (green) and primary tumour samples (yellow, *n* = 47). Black triangles pointing upwards denote an increase in signature contribution in primary tumours; triangles pointing downwards denote a decrease. The centerline of boxes depicts the median values; the bottom and top box edges correspond to the first and third quartiles. The upper and lower whiskers extend from the hinges to the largest and smallest values, respectively, no further than 1.5* the inter-quartile range. Two-sided Wilcoxon signed-rank test *p*-values are displayed (not adjusted for multiple comparisons). Only signatures with a significant change are shown. Source data are provided as a Source Data file.

immune scores, cytotoxicity and infiltration of CD8+/CD4 + T cells (Supplementary Figs. 4, 5). This could indicate promise for effective immune checkpoint blockade for this subset of tumours, which will require further validation in future clinical trials.

We identified an A-T rich hexanucleotide motif genetic scar (SBS41) resembling that induced by colibactin, which has not been described extensively in this cancer (Fig. 2a, Supplementary Fig. 2a–c). The colibactin signature has been predominately reported in colorectal cancers, where it was suggested to be linked with genotoxins

originating from pks+ strains of *E.coli* during tumour progression[21,22]. *E.coli* has been reported to form part of the microbiota in Barrett Oesophagus and OAC, and not of normal oesophagus[23]. To confirm this, we investigated evidence of ID18 exposure in the same samples, as this indel signature has been confidently linked with colibactin toxicity. A total of 18 samples presented evidence of both SBS41 and ID18 with contributions above 5% spread across all disease stages (Supplementary Table 4), therefore suggesting a possible rare contribution of colibactin-induced stress to OAC tumour development.

Additionally, we uncovered a signature of platinum treatment (SBS35) in primary tumours, which is expected given that the majority of these tumours have been sequenced from the surgical resection specimen after treatment with chemotherapeutic agents and platinum is the backbone of treatment regimens. Indeed, this signature was increased specifically in chemotherapy treated samples (Supplementary Fig. 2d) and likely reflects the mutagenic effects of this therapy[24]. While SBS35 is a more generic platinum signature, specific mutational scars left by cisplatin or capecitabine/5-FU treatment have been defined recently[24]. However, we have not found evidence of such signatures in our cohort. 46 patients in the cohort have been treated with carboplatin/cisplatin, but this was almost always in combination with capecitabine/5-FU or other therapies. Hence, it may be that the carboplatin-specific signal (SBS31) gets drowned out by other more prevalent processes. When it comes to the 5-FU signal, we are not able to distinguish it from the pre-treatment SBS17b process which dominates OACs, as the two signatures are essentially identical. It is clear that the SBS17b signature in this cancer is a stand-alone process that acts as an early trigger of OAC, present already in Barrett Oesophagus, and is independent of 5-FU therapy. This does not exclude some common mechanisms to the two processes that we have yet to disentangle. In chemotherapy treated samples this signal may originate both from the original risk factor as well as the 5-FU therapy, but we are unable to distinguish the two.

## Dynamics of mutational processes across disease stages

We found evidence for many mutational processes acting very early on in tumour evolution such that they are already present in pe-cancerous Barrett Oesophagus, especially SBS17a/b and the ageing-linked signatures SBS1, 5 and 40 (Fig. 2b). Despite the fact that the Barrett samples encompassed an entire spectrum from non-dysplastic non-progressors and pre-progressors to low/high grade dysplasia, intramucosal carcinoma and Barrett Oesophagus adjacent to the cancer, the signature prevalence did not differ significantly across these categories (Supplementary Fig. 6a), which were also clearly genomically distinct from the primary cancer, with copy number and ploidy profiles in line with those expected in pre-malignant disease (Supplementary Fig. 6b, c). Thus, the presence of most signatures early in Barrett Oesophagus samples is unlikely to be due to a confounding effect of malignancy already existing in some of the more advanced cases given the heterogeneity of this cohort, but rather due to most of these processes acting very early on before tumour establishment.

Signatures SBS28 and SBS35 are scarcely visible in Barrett Oesophagus (two and one samples, respectively, with an exposure >5%) but they are clearly visible in primary tumours suggesting that they are primarily operative at the stage of invasive disease. This is expected given that SBS35 is linked with platinum treatment, while SBS28 has been linked with polymerase epsilon mutations but also shares similarities with SBS17b and thus could also explain a noisier or imperfectly deconvolved signal in the already established malignancy, possibly also due to 5-FU treatment. We also observed a general increase in the contribution from APOBEC-linked mutagenesis (SBS2) and HR/MMR deficiencies (SBS3 and SBS44) in primary tumours, followed by decrease in metastases, while the SBS17 processes tended to rise further in metastases (Fig. 2b, Supplementary Fig. 7a). Most changes were independent of the treatment status of the samples (Supplementary Figs. 8, 9). While treatment-related signatures may be expected to be enriched in metastases given enough time for a complete clonal expansion, the observed SBS17a/b increase in metastases was similar between naïve and treated cases and thus most likely due to the original unknown trigger. However, we cannot exclude some contribution from the 5-FU treatment, which is difficult to disentangle due to the aforementioned separation problem between the SBS17-OAC-specific process and the 5-FU signatures, which are highly similar[25]. Nevertheless, it is worth noting that most of the primary tumour and metastatic samples analysed were not originating from the same patients, and for the cases where matched samples were available the increasing/decreasing trends were less clear. Larger cohorts of matched primary-metastatic cases will be needed in the future to further investigate these patterns.

Ageing-associated mutational events (SBS1 and 5) generally appeared as a continuous background contribution that stabilises in primary tumours and metastases and shows a relative decrease in prevalence compared to other signatures. However, SBS40, also thought to be linked with ageing, increased from pre-malignant to advanced disease both in relative and absolute counts (Fig. 2b, Supplementary Fig. 7a) – suggesting that the derivation of this mutational process is different and may have a higher impact in this disease than previously appreciated. The dynamics of such ageing-related processes could also be linked with ageing-induced mutagenic drift observed during Barrett Oesophagus development, which can be present years prior to cancer initiation[26,27].

Comparing matched samples of Barrett Oesophagus and primary tumours from the same individuals ($n = 51$) further corroborated our previous findings: APOBEC mutagenesis, HR impairment, the SBS40 process and the platinum signatures increase in prevalence with disease progression (Fig. 2c, Supplementary Fig. 7b). The ageing signatures 1 and 5 and the oxidative stress signature S18 decrease in importance, but continue to contribute mutations in the primary tumour (SBS1) or stabilise (SBS5, SBS18). It should be noted, however, that the prevalence of SBS2, 3 and 35 in the matched cases was most often below 5%, and thus below the threshold on which we can confidently call a mutational process as significantly acting in the tumour (see Methods). Thus, after applying this threshold significant changes were only observed in the SBS18 and ageing processes.

## Nucleosome periodicity of mutations across disease stages

Mutation rates along the genome are highly variable and influenced by several chromatin features. SBS17-associated T > G and T > C substitutions were enriched on the untranscribed and lagging strands, confirming previous studies[28], and this was consistent across the disease spectrum from Barrett to primary and metastatic disease (Supplementary Fig. 10). The rate of mutations in nucleosome covered DNA followed a periodic pattern with maximum power period (MP) of ~10.3 with increased mutation rate when the minor groove faces the nucleosome, consistent with previous reports[13]. This periodic pattern was similar across stages, with significant signal-to-noise ratios (SNRs) that increased in primary tumours compared to Barrett Oesophagus (Supplementary Figs. 11, 12).

Periodic patterns were most prominently observed for SBS17a/b-linked mutations, with maximum power periods of ~10.3 and 10.15, respectively, across all disease stages (Supplementary Fig. 13a, b). SBS18-linked mutations also displayed periodicity in Barrett Oesophagus and primary tumours, but not in metastases – although this may be due to lower exposure in advanced cancers (Supplementary Fig. 13c).

The results corroborate previous evidence that mutations in nucleosome covered DNA follow a 10.3 bp periodicity pattern in oesophageal cancer, and from another perspective demonstrate that this periodicity is already present in mutations in Barrett Oesophagus, indicating that this signal is essentially the same across stages. This mutation periodicity is especially clear in SBS17 mutations. The reason for this periodicity has been attributed to differential DNA repair processes in stretches of DNA with the minor groove facing histones and away from them. In particular, SBS17 linked to 5-FU treatment may be caused by alterations in the pool of nucleotides available for DNA synthesis[29], which could lead to misincorporation of nucleotides during DNA replication. These misincorporated nucleotides could be, at least in part, repaired by Base Excision Repair (BER), which we have previously shown follows a periodic pattern[13]. Mutation periodicity for

SBS17 not linked to 5-FU could have a similar explanation, although for the moment, as the aetiology of this signature is not clear we cannot do more than speculate.

## Risk factors and clinical associations

The risk factors that sustain OAC mutagenesis have remained poorly characterised to date due to the scarcity of high quality matched clinical annotations, and the aetiology of the SBS17 processes which dominate this disease remains unknown. To get further insights into the potential mutagenic triggers of this cancer, we correlated mutational signatures observed in Barrett Oesophagus and primary tumours with reported environmental exposures. These annotations were not available for metastatic samples.

No marked strong correlations ($p < 0.01$ or lower) were observed with alcohol consumption or non-steroidal anti-inflammatory drug (NSAID) usage (Supplementary Fig. 14). Current/past smokers presented increased levels of SBS17a/b compared to never smokers in the primary tumours, but not in Barrett Oesophagus (Supplementary Fig. 15). Mutational loads did not vary by smoking status at any premalignant or cancer stage (Supplementary Fig. 16), and DDR associated mutations were also broadly similar (Supplementary Fig. 17), with a marginal depletion of SNVs affecting genes involved in direct repair in never smokers (Fisher's exact test adjusted $p < 0.05$, odds ratio = 0.15). Overall, no strong signals of a protective effect from mutagenesis appeared to be present in never smokers in precancerous stages. However, we did observe a significantly lower fraction of smokers in the non-dysplastic Barrett Oesophagus patients that do not go on to progress to cancer compared to all the other categories (Fisher's exact test $p = 0.009$, odds ratio = 0.24, Supplementary Fig. 18). This is in keeping with smoking being a known risk factor for progression to OAC[30].

No link was found between any signature and PPI/acid suppressant usage. However, the majority of the cohort would have been expected to use these drugs at some stage before or after diagnosis, so any such correlations may be difficult to discern. Furthermore, data on reflux symptoms is poorly recorded, further limiting the insights on acid reflux association with mutational processes.

## Cancer drivers and mutational signature impact

To check whether any of the observed mutational signatures shaping disease stages in OAC might be linked with specific driver events, we surveyed the mutational and copy number landscape of key drivers of OAC as described by ref. 31. Overall, the top drivers remain fairly consistent from preneoplastic samples to primaries and metastases, with a higher prevalence of *TP53* mutations in primary tumours and metastases, as expected (Fig. 3a–c, left). Genomic changes affecting several driver genes were associated with increases or decreases in the prevalence of the SBS17 signatures as well as ageing-related (SBS5/40), BER (SBS30) and MMRD signatures (SBS44) (Wilcoxon rank-sum tests, Fig. 3a–c, right). None of the associations where significant after multiple testing correction in Barrett Oesophagus, possibly due to reduced driver frequency which reduces statistical power. However, several such events remained significant in primary tumours as well as metastases (Fig. 3b, c, right).

Most notably, *TP53* alterations were linked with an increased prevalence of SBS17a/b in primary tumours, while *MDM2* changes showed the opposite trend, pointing towards a consistent association across the same pathway (Fig. 3b, right). In contrast, SBS17b boosts alone appeared strongly associated with *MUC6* and *AXIN1* events in metastases (Fig. 3c, right). Interestingly, *CDK6* mutations appeared linked with SBS17 mutagenesis both in Barrett Oesophagus as well as metastases (Fig. 3a, right). *CDK6* is a cyclin dependent kinase which drives the cell cycle through pRB inactivation in G1, and an emerging target in cancer together with *CDK4*. Indeed, when applying our previously developed signature of proliferation/cell cycle arrest to this cohort[32], we observed a significant increase in proliferation capacity in samples with SBS17 contributions above 5% (Supplementary Fig. 19), suggesting that SBS17 mutagenesis may be enhanced in faster growing, more aggressive tumours enabled through *CDK6* activation. Finally, changes in the gene *ACVR2A*, a transmembrane receptor linked with TGFβ signalling, were correlated with a prominent increase in MMR deficiency (SBS44), which may pinpoint to linked mechanisms of immune evasion.

## Mutational processes driving invasive disease

Given the observed fluctuations in mutational scars between Barrett Oesophagus, primary tumours and metastases, it is reasonable to expect that certain mutational processes might contribute to the progression from pre-malignant to invasive disease. Within an individual disease stage, we observed various combinations of mutagenic processes acting in the genomes (Supplementary Fig. 20), some of which were common between stages, such as the joint presence of SBS17a/b and SBS40, and some of which were unique, e.g., SBS41 and all ageing-linked signatures were only observed to co-occur in primary tumours. To make sense of this complexity, we asked whether we could prioritise signatures that can help us distinguish between Barrett Oesophagus, primary tumours and metastases. In other words, could certain mutagenic patterns predict disease progression? To this end, we employed gradient boosting and random forest classifiers to distinguish between cancer stages based on the mutational footprint alone (see Methods).

When considering the overall signature contributions in each cancer stage, the models distinguishing Barrett Oesophagus from primary tumour genomes had performances of 84–86% AUC (Fig. 4a), suggesting that the combined mutational scars left in the genome during the course of this malignancy can help distinguish disease boundaries remarkably well. The APOBEC mutagenesis signature was ranked as the most predictive of primary tumour development, followed by the ageing-linked SBS40 and SBS1, suggesting they may be more important in driving the malignant transformation of pre-neoplastic lesions (Fig. 4b). Interestingly, APOBEC3A has been recently shown to preferentially mutate VpC and TpC hotspots in cancer drivers such as *PIK3CA* and *KRAS*[33], which we find are specifically selected in primary tumours compared to Barrett Oesophagus or metastases (Fig. 4c, d). Indeed, *PIK3CA* mutant cancers showed an increased SBS2 prevalence in our cohort (Wilcoxon rank-sum test $p = 0.005$). Although our analysis indicates *PIK3CA* and *KRAS* mutations as conferring a selective advantage at primary tumour stage, they are not exclusive to primary tumours and in fact are also more rarely found in Barrett Oesophagus (2 cases with *KRAS* mutations, 3 with *PIK3CA* mutations). Overall, this may indicate that the increased APOBEC mutagenesis may facilitate the acquiring of key drivers for OAC progression, which are likely important for enabling the establishment of the tumour although they are not linked with survival outcomes (Supplementary Figs. 21, 22). This signature association with the primary tumour stage was further corroborated by a multinomial regression analysis (Fig. 4e). The ageing signature S1 appeared most specific to Barrett Oesophagus cases, which is not surprising given that it is the primary source of mutations in healthy tissues.

Interestingly, it emerged from the model that the clonality of the mutations had a strong contribution to distinguishing between cancer stages (Fig. 4b). This was despite the fact that Barrett Oesophagus and primary tumour samples had similar purities both when fully clonal as well as when presenting subclonally (Supplementary Fig. 23). Thus, the differences in clonality picked up by the model are unlikely to simply reflect normal cell contamination in Barrett Oesophagus but rather a genuine effect of the clonal or subclonal action of specific mutational processes. As a result, we built a second gradient boost classifier that would enable us to highlight processes that act subclonally or later in

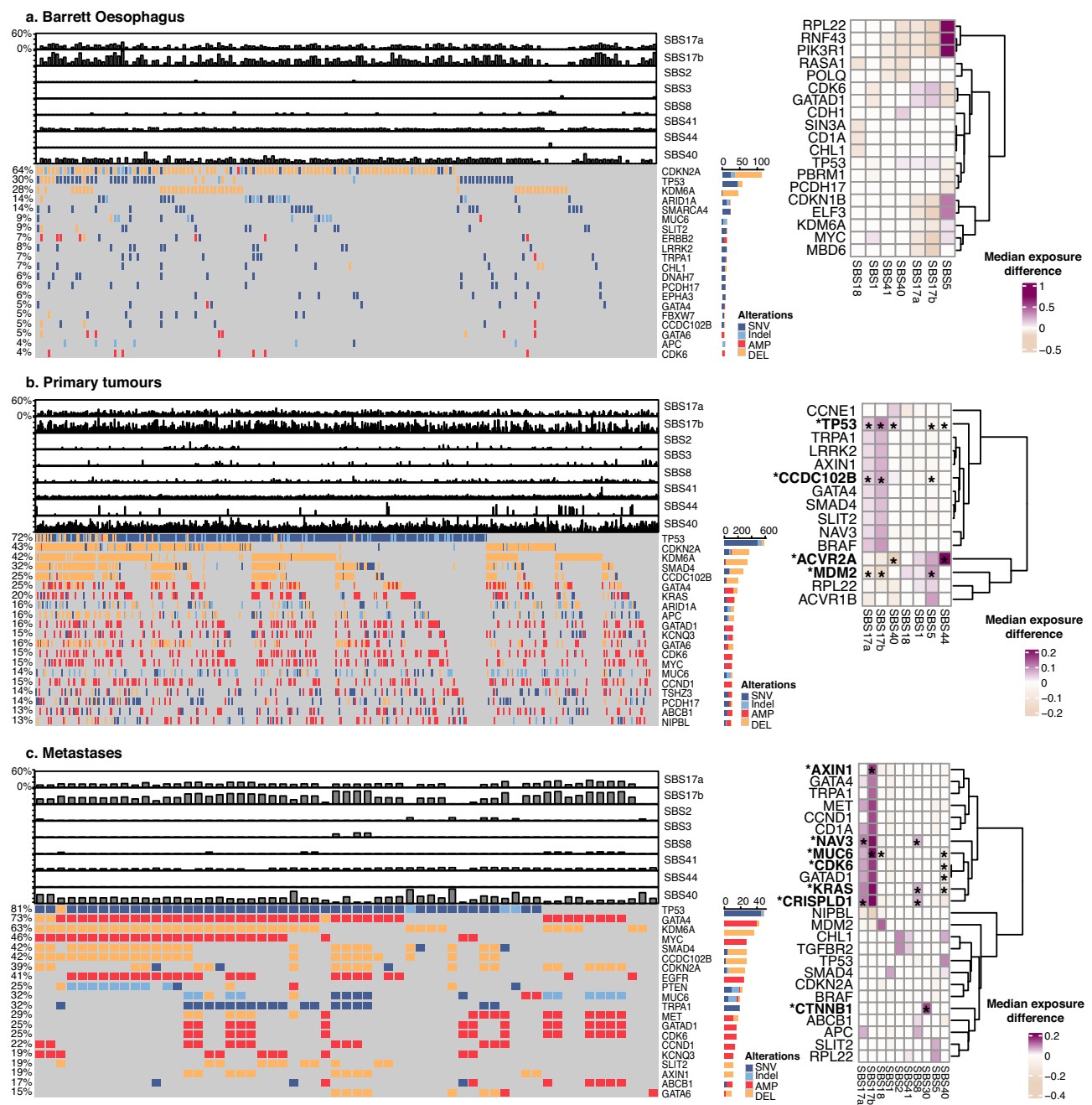

**Fig. 3 | Cancer drivers across disease stages and mutational signature prevalence.** The landscape of cancer driver events and their association with mutational signatures are depicted for (**a**) Barrett Oesophagus (*n* = 161), (**b**) primary tumours (*n* = 777) and (**c**) metastatic samples (*n* = 59). Left: The mutational and copy number alteration landscape of OAC drivers is shown in parallel with the contribution of key mutational signatures (top bar plots). The fraction of samples with alterations in a specific gene is shown on the left and in the bar plots on the right. Differennt alteration types are denoted with different colours. SNV = single nucleotide variant, AMP = amplification, DEL = deletion. Right: Heat maps depicting the increase (values >0) or decrease (values <0) in mutational signature prevalence in samples harbouring mutations or copy number changes in specific genes. The colour gradient indicates the median change in exposure compared to wild type. Only significant changes of >10% (in either direction) are depicted with Wilcoxon rank-sum two-sided test *p*-value <0.05. Stars indicate associations that are still significant after FDR multiple testing correction. Source data are provided as a Source Data file (including *p*-values).

evolution in a stage-specific manner, which had an accuracy of 86% (Fig. 4f). This model confirmed the key signals from the previous analysis, but shed further light on the fact that the APOBEC and SBS41 mutations that appear as a distinct signature in primary tumours are accumulated clonally later (APOBEC) and earlier (SBS41) in evolution, respectively. Furthermore, SBS17b clonal mutations that accumulated later in evolution emerged as the most specific for Barrett Oesophagus genomes. The SBS17 signatures emerged amongst the top patterns linked with Barrett Oesophagus also when predicting this precancerous stage from primary tumours using both mutational and indel signatures while removing signature contributions of <5%, although at a slightly lower performance of 83% AUC (Supplementary Fig. 24). Thus, while the APOBEC signature appears quite specifically increased in primary tumours, its overall contributions are fairly low, while SBS17 contributions are markedly high in pre-neoplasia. In addition, indel signatures ID1 and ID2, linked with slippage during DNA replication, also ranked highly in distinguishing primary tumours (Supplementary Fig. 24).

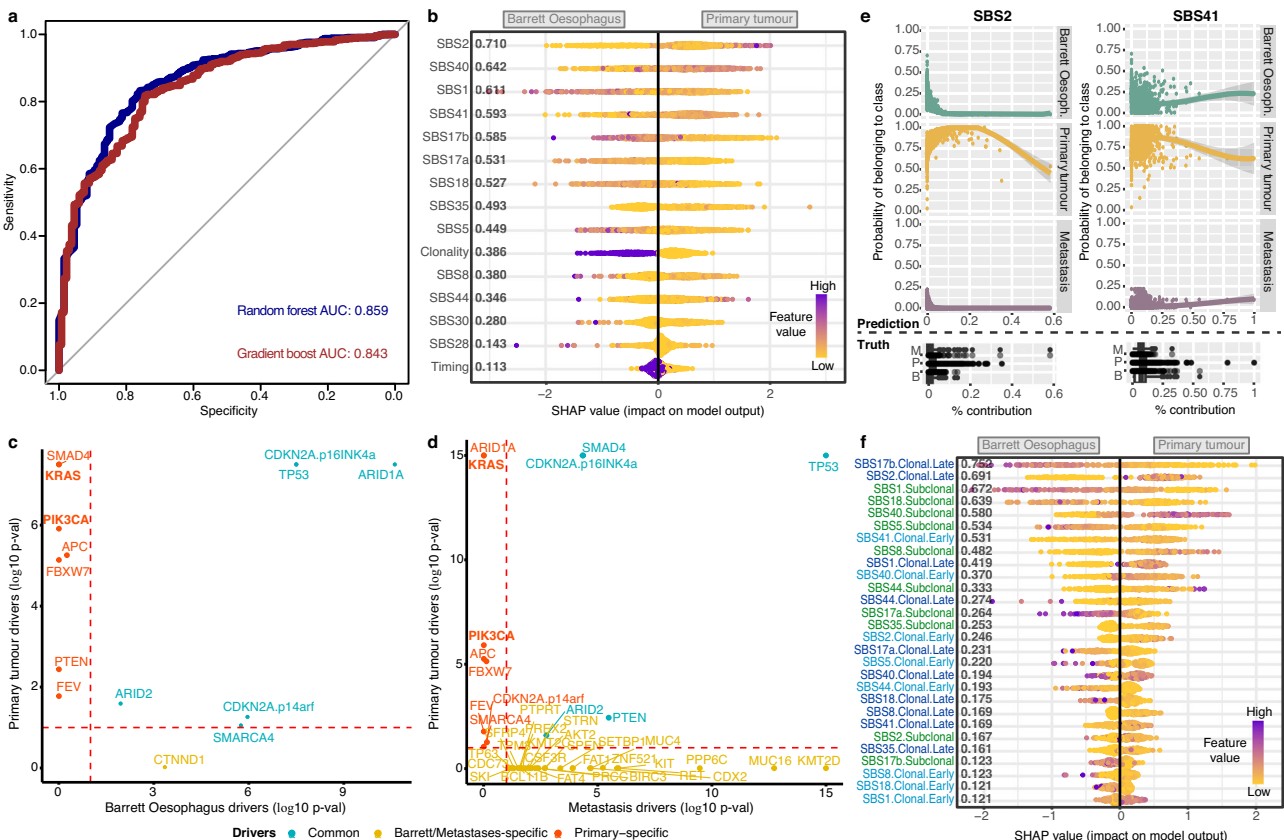

**Fig. 4 | Mutational processes and driver events distinguishing primary tumours from Barrett Oesophagus genomes. a** Performance of the gradient boosting and random forest signature-based classifiers in distinguishing between Barrett Oesophagus and primary tumours. The area under the curve (AUC) is indicated for either model. **b** Output of xgboost model distinguishing Barrett Oesophagus from primary tumours based on overall signature prevalence, while accounting for clonality and timing. Features are ordered according to their ranking in the model (top ranking features first). Every dot is a sample and the colour corresponds to the signature contribution in that sample, ranging from purple (highest contribution of the respective signature across the cohort) to yellow (lowest contribution of the respective signature). For 'clonality'/'timing' purple denotes clonal/early and yellow denotes subclonal/late. **c** Genes positively selected in primary tumours versus Barrett Oesophagus. Genes commonly positively selected in all tumours are highlighted in blue. Genes positively selected only in the primary tumour group are highlighted in red, only in Barrett Oesophagus in yellow. *KRAS* and *PIK3CA* mutational events, specific to primary tumours, are highlighted in bold. The log likelihood-ratio test p-values are reported, adjusted for multiple testing using the Benjamini-Hochberg method. **d** Genes positively selected in primary tumours

versus metastases are shown similarly as in (**c**). The log likelihood-ratio test *p*-values are reported, adjusted for multiple testing (Benjamini-Hochberg method). **e** Multinomial regression classifier results distinguishing Barrett Oesophagus, primary tumours and metastases based on signature prevalence. The predictive power of SBS 2 and 41 in distinguishing primary tumours is exemplified. The top panel shows the predicted disease stage depending on increasing mutational signature prevalence. The bottom panel shows the true distribution of mutational contributions for the selected signatures among three stages, with the centerline of boxes depicting the median exposure, the bottom and top box the first and third quartiles, and upper and lower whiskers extending from the hinges to the largest and smallest values, respectively, no further than 1.5* the inter-quartile range. (M = metastasis; P = primary tumour; B = Barrett Oesophagus). The curves in the prediction model were fitted with a loess function (shaded areas depict the 95% confidence interval). **f** Output of xgboost model distinguishing Barrett Oesophagus from primary tumours based on detailed signature contributions split by clonality and timing. Early clonal events ar depicted in light blue, late clonal events in dark blue and subclonal events in green. The individual dots are coloured as described in (**b**). Source data are provided as a Source Data file.

Our power to detect signature differences when comparing primary tumours to metastases was reduced due to the smaller size of the metastatic cohort (despite an accuracy of 92%), but we could observe a prominent contribution from a subclonal signature SBS17b in metastases (Supplementary Fig. 25).

While we are not proposing these classifiers for clinical application, this analysis does suggest that there are distinct contributions of mutational processes over a lifetime of a tumour which are prevalent enough to be somewhat predictable.

## DNA repair pathway dysregulation modulates OAC progression

We next investigated how DDR regulation might contribute to shaping the mutational landscape of this disease. First, we asked to what extent the different pathways involved in repairing DNA damage are altered via SNVs, indels or copy number changes in the cohort. We surveyed such changes across >400 genes acting in 13 DDR-related pathways as

described in ref. 34. Among the most frequently altered pathways were BER, nucleotide excision repair (NER), HR, translesion synthesis (TLS), Fanconi Anaemia, mismatch repair (MMR) and non-homologous end joining (NHEJ), particularly based on the frequency of deletions which affected more or nearly half the patients, while sparser events affected other pathways (Fig. 5a). We also confirmed that changes in these pathways were linked with an increased SNV or indel signature contribution of the same expected mutagen in the cohort (Fig. 5b). However, considering the broad prevalence of DDR pathway alterations in the cohort, we could observe the mutational footprint left by deficiencies in these processes to be relatively low – which suggests a remarkable robustness encoded in these pathways.

To investigate whether the tumours that present distinct DDR alterations also display downstream transcriptional changes, we employed matched RNA-seq data that we had available for 203 OAC cases. When investigating the expression of the genes involved in the

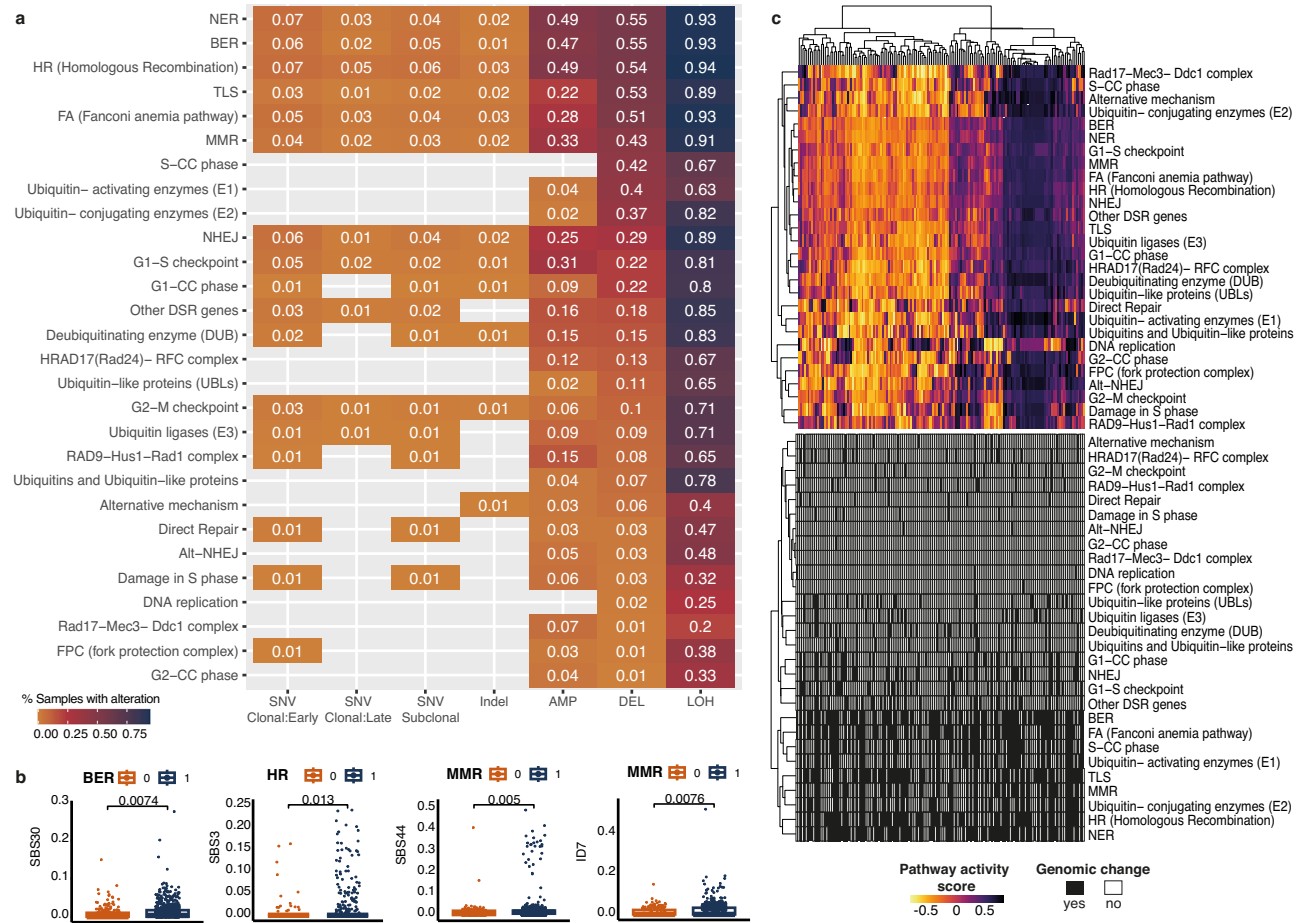

**Fig. 5 | DNA damage repair drivers and downstream transcriptional activity.**
**a** SNVs, indels and copy number changes affecting DDR pathways. The heat map highlights the fraction of primary tumour samples ($n = 777$) with a specific type of change in each pathway and rows are ordered by decreasing fractions of deletions in pathway. Increasing fractions of alterations are denoted by a colour gradient from orange to purple. AMP = amplification, DEL = deletion, LOH = loss of heterozygosity. **b** SNV and indel signature contributions compared between samples with (1) and without (0) a specific DDR defect ($n = 771$ independent samples with BER defects and $n = 226$ without; $n = 773$ independent samples with HR defects and $n = 224$ without; $n = 769$ independent samples with MMR defects and $n = 228$ without). The centerline of boxes depicts the median values; the bottom and top

box edges correspond to the first and third quartiles. The upper and lower whiskers extend from the hinges to the largest and smallest values, respectively, no further than 1.5* the inter-quartile range. Two-sided Wilcoxon signed-rank test $p$-values are displayed. All plots confirm increased signature contributions when the pathway is mutated. **c** Top heat map: Sample by sample activity in every DDR-related pathway as measured from expression of genes implicated in the pathway using GSVA, displayed for $n = 203$ profiled primary tumour samples. The colour gradient varies according to the pathway activity score. Bottom heat map: Prevalence of SNV/indel and copy number changes for the same samples in the respective pathway (black = altered, white = non-altered). Source data are provided as a Source Data file.

different DDR pathways (see Methods), we observed a good coordination across most pathways, with samples splitting into three broad patterns of relative upregulation, downregulation, or moderate activity across most DDR pathways concomitantly (Fig. 5c, top). We did not observe any clear clustering of pathway transcriptional activity by mutational patterns in the respective pathways, which reflects a complex relationship between genome scars and downstream gene/protein-level activity (Fig. 5c, bottom). Overall, these findings suggest that the DDR pathways appear fairly resilient in OAC.

**Tumour clonal heterogeneity uncovers SBS17 mutagenic shifts**
To further understand how mutational processes shape evolutionary trajectories in OAC, we investigated the timing of mutation accumulation due to the different neoplastic processes identified in the cohort. We identified frequent subclonal events (~51% of samples) where mutational pressures change (Fig. 6a). Most of these changes were consistent across tumour stages, with the exception of SBS18 and SBS5, which increased only in Barrett subclones, and decreased in primary tumour and metastasis subclones. Several processes,

including SBS17 and SBS18, presented clear subclonal changes, whereas others, like SBS30 or SBS28, appeared stable on average. The most notable change was a subclonal decrease in SBS17a/b mutations, corroborating the findings from the PCAWG consortium study in primary tumours[9]. The lower subclonal exposure was observed across the stages, from Barrett Oesophagus to primary tumours and metastases, but with a slight progressive pattern. These were by far the most dominant signals of dynamic shift observed during OAC evolution. Thus, we focused on exploring the genetic and pathway dependencies of SBS17 more broadly in the cohort in order to shed clarity into potential consequences of the clonal dominance of SBS17.

Multiple cancer drivers involved in chromatin remodelling and transcriptional control, including *SMARCA4*, *KMT2D* and *ARID2*, were positively selected in samples with abundant SBS17 signals (Fig. 6b).

Tumours with SBS17b exposure displayed increased ploidy and chromosomal instability, as well as higher DDR activity, telomere maintenance, cell cycle control and angiogenesis (Fig. 6c). Furthermore, samples where the SBS17 was more prevalent subclonally

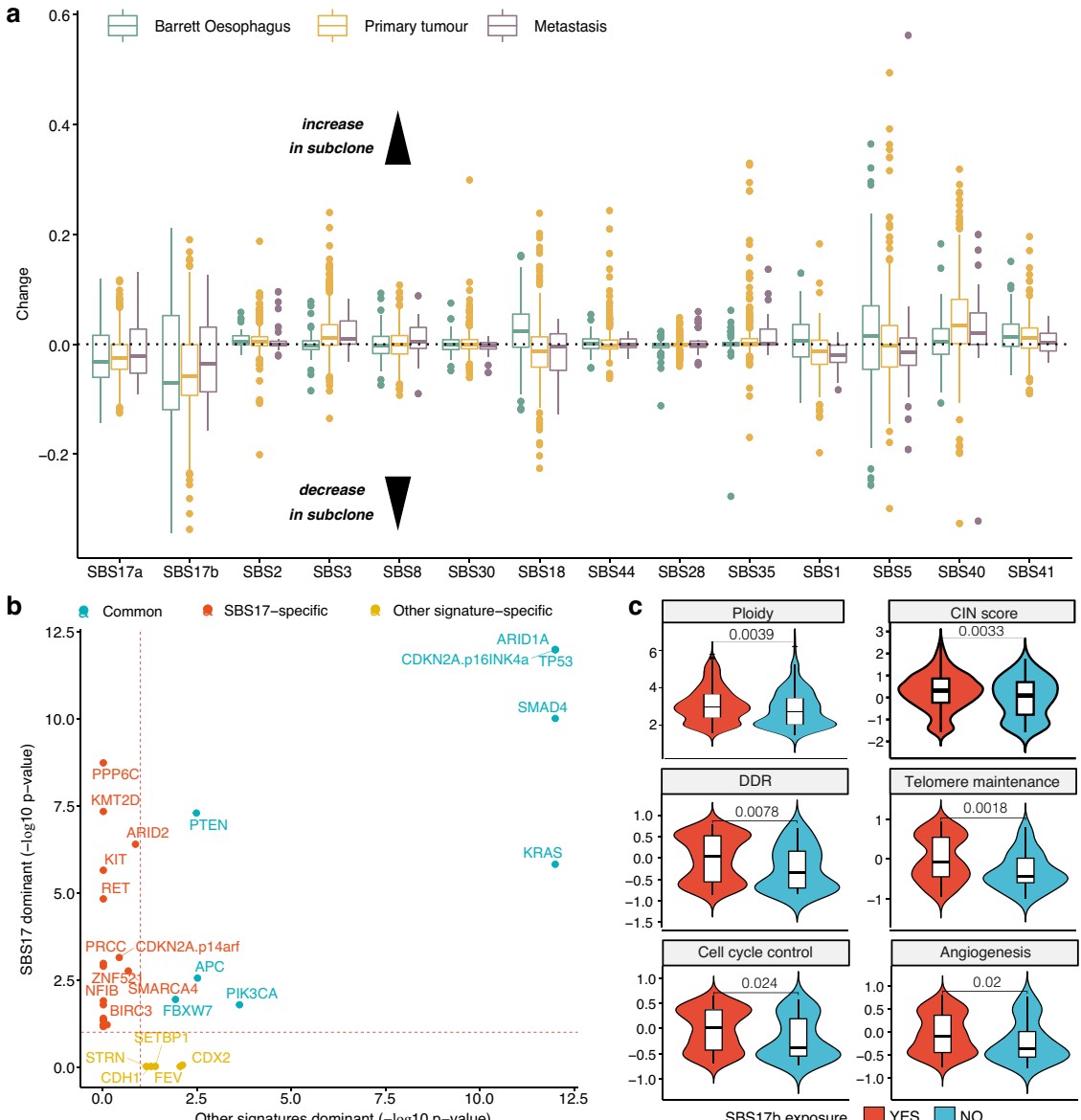

**Fig. 6 | Tumour clonal heterogeneity reveals widespread SBS17 shifts that correlate with changes in cellular phenotypes. a** Changes in signature exposure in tumour subclones. Values below 0 indicate a decrease in signature exposure in the sublones, values above 0 an increase. Signatures SBS17a and b are the only ones showing a dominant decrease across Barrett ($n = 89$, green), primary ($n = 512$, yellow) and metastatic ($n = 38$, purple) stages. Box boundaries represent first and third quartiles, centerline indicates median values. The upper and lower whiskers extend from the hinges to the largest and smallest values, respectively, no further than 1.5* the inter-quartile range. Outlier points are plotted individually. **b** Positively selected genes in primary tumours with a dominant SBS17 signature versus the ones positively selected in tumours with other dominant signatures. Genes commonly positively selected in both categories are highlighted in blue. Genes positively selected only in the SBS17 dominant group are highlighted in red. **c** The presence of SBS17b is associated with an increase in ploidy and chromosomal instability (CIN), as well as higher activity of telomere maintenance, DNA damage repair (DDR), cell cycle control and angiogenesis pathways. The YES category (red) denotes samples with SBS17b exposure >5% ($n = 831$ for genomic measurements; $n = 167$ for transcriptional hallmarks), while the NO category (blue) refers to exposures <=5% ($n = 164$ for genomic measurements; $n = 36$ for transcriptional hallmarks). Box boundaries represent first and third quartiles, centerline indicates median values. The upper and lower whiskers extend from the hinges to the largest and smallest values, respectively, no further than 1.5* the inter-quartile range. Two-sided Wilcoxon signed-rank test $p$-values are displayed. Source data are provided as a Source Data file.

harboured a functional, wild type p53 (Fisher's exact test $p = 0.0004$, 1.9-fold enrichment) and showed a slight reduction of CD8+/CD4+ T cell infiltration (Wilcoxon rank-sum test $p < 0.05$), as inferred from the expression of cell type-specific markers using ConsensusTME (Supplementary Fig. 26). Wild type p53 often marks slower growing tumours, which could explain the decreased immune responses observed. This observation is consistent with what we would expect to see in samples where SBS17 is not a clonal process, according to our previous analyses which showed that a strong SBS17 prevalence links with higher proliferation.

## Clinical relevance of mutational signatures

Finally, we sought to investigate whether any mutational processes present links with outcomes observed in the clinic for OAC patients. Remarkably, the mutational signature linked with BER impairment, SBS30, was the most prognostic in our cohort, even after accounting for confounding factors such as age, gender, stage (Fig. 7a, Supplementary Table 5). Patients showing any evidence for BER deficiency in their tumours (>5%, cut-off determined by robustness tests of mutation callers - see Methods) had a worse overall survival (Fig. 7a), suggesting a potential prognostic utility for this signature in the clinic.

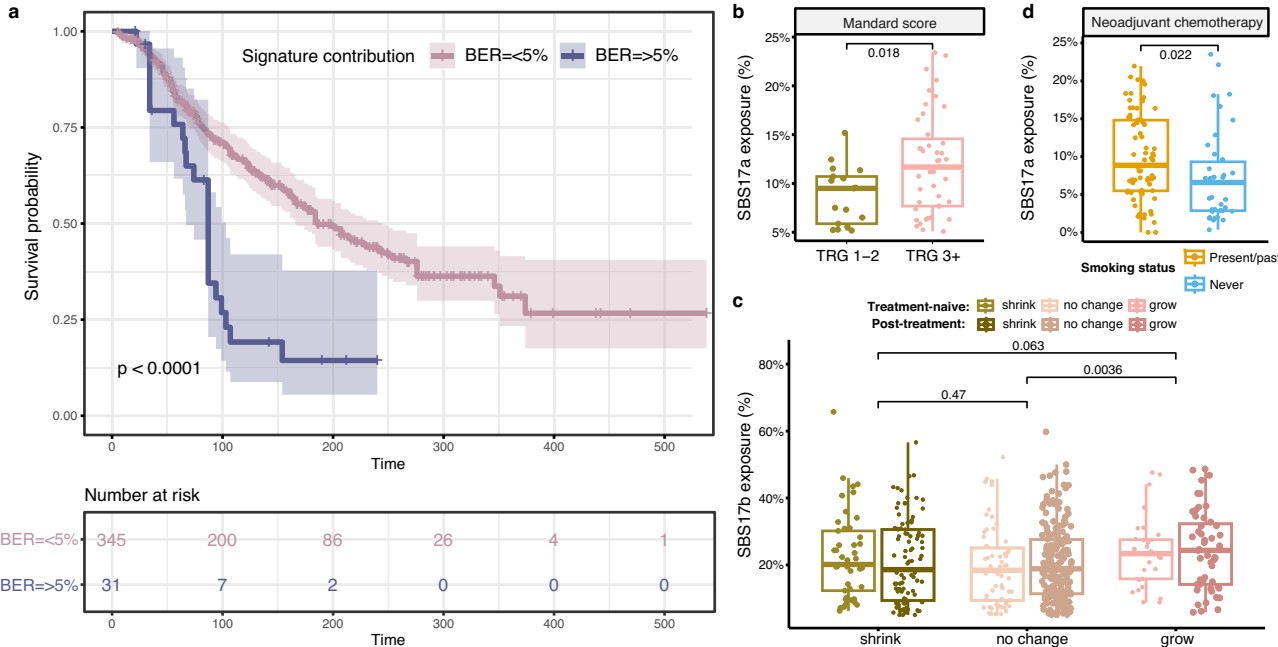

**Fig. 7 | Associations between mutational signatures and clinical outcomes.**
**a** Patients with a BER signature prevalence >5% have a significantly worse overall survival outcome, as depicted by Cox proportional hazards analysis (Cox model log-rank $p = 7.3e-06$). The shaded areas depict the 95% confidence intervals. The log-rank test p-value is reported. The number of patients at risk at various time intervals are shown in the table below. **b** Prevalence of the SBS17a signature in treatment naïve tumours, compared between patients with Mandard scores TRG 1-2 ($n = 17$) versus TRG 3 or higher (TRG 3 +, $n = 43$). The centerline of boxes depicts the median values; the bottom and top box edges correspond to the first and third quartiles. The upper and lower whiskers extend from the hinges to the largest and smallest values, respectively, no further than 1.5* the inter-quartile range. A two-sided Wilcoxon signed-rank test *p*-value is displayed. **c** Prevalence of the SBS17b signature in tumours that shrunk ($n = 145$), showed no change ($n = 242$) or grew ($n = 83$) after surgery. The change in tumour volume (T stage) was calculated from pre-treatment staging to post-therapy resection pathology staging. The tumours are further split based on whether the sequencing was performed on the treatment-

naïve ($n = 136$) or post-treatment ($n = 334$) sample. No major differences are observed based on treatment status, but an increase in SBS17b is seen in tumours growing after surgery. The groups are coloured according to treatment and volume change. Box boundaries represent first and third quartiles, centerline indicates median values. The upper and lower whiskers extend from the hinges to the largest and smallest values, respectively, no further than 1.5* the inter-quartile range. Two-sided Wilcoxon signed-rank test p-values are displayed. **d** SBS17a signature prevalence in post-treatment tumour samples from non-responders to neoadjuvant chemotherapy (i.e., patients showing stable or progressive disease), compared between present/past smokers ($n = 71$, orange) and never smokers ($n = 38$, blue). Higher SBS17a contributions are observed in smokers. Box boundaries represent first and third quartiles, centerline indicates median values. The upper and lower whiskers extend from the hinges to the largest and smallest values, respectively, no further than 1.5* the inter-quartile range. A two-sided Wilcoxon signed-rank test *p*-value is displayed. Source data are provided as a Source Data file.

SBS17a and SBS17b exposures did not show significant associations with overall survival outcomes (Supplementary Table 6). However, patients with worse tumour regression outcomes, i.e., Mandard TRG 3 or higher, presented increased SBS17a mutagenesis in the tumour before treatment (Fig. 7b, Supplementary Fig. 27). We further assessed tumour progression by growth in tumour volume from pre-treatment staging to post-therapy resection, and detected an increased SBS17b prevalence in tumours that grew after surgery (Fig. 7c). This was observed both in tumours sequenced before as well as after treatment, potentially hinting at an early SBS17 mutagenic link with patient outcomes. When examining poor responders to neoadjuvant chemotherapy, we found that past or present smokers showed an increased SBS17a mutagenesis signal in their tumours after treatment compared to never smokers (Fig. 7d). No differences were observed in individuals presenting complete or partial response to chemotherapy by smoking status (Supplementary Figs. 28, 29). A similar trend was observed in tumours in the context of radiotherapy, but these did not reach statistical significance (Supplementary Figs. 28b, 29b). This, in conjunction with our previous observations that SBS17 signatures tend to be more prevalent in faster proliferating tumours, could indicate a role of this mutational process in conferring more aggressive phenotypes that are also more resistant to standard therapies for OAC. These observations should nevertheless be considered in light of the dominance of stage T3/4 tumours in our cohort (76% of cases), which limits the chance to observe progressive disease.

## Discussion

This present study of mutational processes during the course of OAC development from pre-cancerous stages to advanced spread provides an extensive description of the dynamics of mutational events during tumorigenesis in this disease. Building on our knowledge of mutational signatures operative in OAC tumours[16], we have elucidated details about the temporal behaviour of mutational processes during the evolution of OAC, summarised in Fig. 8.

We have characterised and compared the landscape of mutational processes at each stage of OAC carcinogenesis. We showed that OAC evolution is marked by frequent mutational signature changes in relation to the clonal composition of the tumour. The dominant SBS17b/a process appears to be triggered early in preneoplastic stages and is accompanied by increased copy number instability, DDR and telomerase activity, suggesting a role in promoting tumour progression. We confirmed that the nucleosome periodicity of this mutational process[13] is maintained across cancer stages, and found that chromatin remodellers such as *SMARCA4*, *KMT2D* and *ARID2* appear to be selected for in the presence of this signature. Interestingly, SBS17 was prominently clonal and linked with genomic and transcriptional markers of highly proliferating, more aggressive tumours, including *CDK6* mutations. Potentially this is most important in conferring a proliferative advantage in the incipient stages of cancer, since association with *CDK6* are observed in Barrett Oesophagus but not in primary tumours, and in metastases which could be due to the fact that the latter are

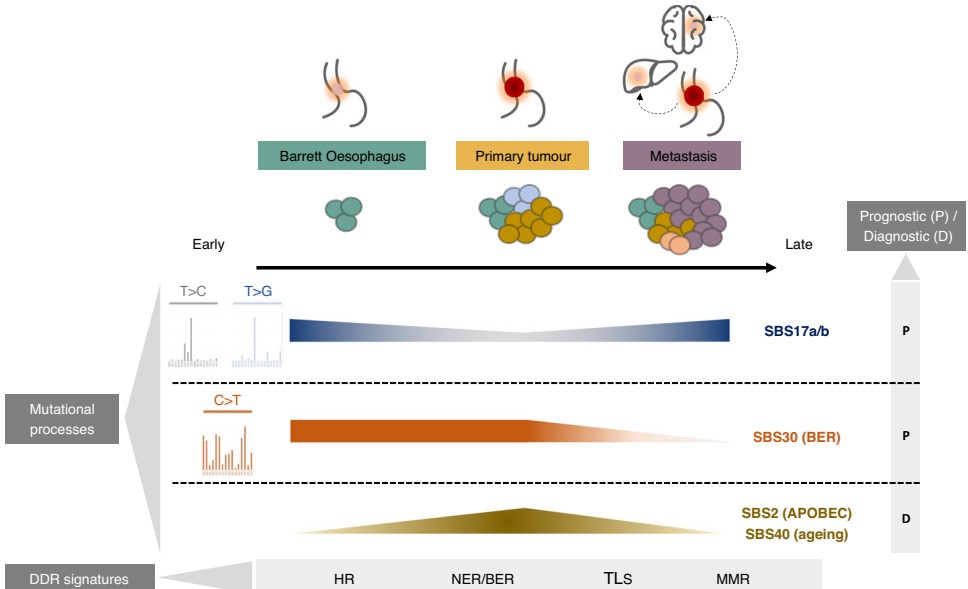

**Fig. 8 | Key genomic signatures underlying distinct exposures, expansion and outcomes during OAC evolution from pre-cancerous to advanced disease.** SBS17a/b processes show a relative decrease in the primary tumours compared to Barrett and metastasis cases, and can be used as markers of transformation early within Barrett Oesophagus. Frequent subclonal changes appear for this signature. SBS30 appears elevated in Barrett Oesophagus and primary tumours, and is linked with worse survival. Several signatures including SBS2 and SBS40 are most highly represented in primary tumours and can be used to distinguish this stage. DDR signatures specifically elevated at distinct points during OAC evolution are also highlighted. Links between signatures and prognosis/diagnosis are highlighted: SBS17a/b presence marks increased proliferation and progression after treatment, SBS30 is associated with worse prognosis, while SBS2 and SBS40 signatures could be used to specifically distinguish (diagnose) primary tumours. P = prognosis, D = diagnosis; HR = homologous recombination; NER = nucleotide excision repair; BER = base excision repair; TLS = translesion synthesis; MMR = mismatch repair.

often seeded early during OAC development[35]. Since CDK4/6 inhibitors have shown promise in preclinical studies in OAC[36], it is possible such inhibitors may be most effectively targeted at the patient segment showing high SBS17 levels in advanced metastatic disease. Such associations by no means suggest any causal link between these drivers and mutational processes, but their co-occurrence warrants further investigation and may inform patient stratification for certain therapeutic regimens.

Observed subclonal decreases in SBS17 intensity in the course of the disease may be due to inflammation-triggering processes becoming better controlled through treatment, to confounding effects from the termination of chemotherapy, or simply to changes in environmental or cell-intrinsic triggers whereby other mutational processes take over. The higher subclonal prevalence of SBS17b in metastases might suggest that further mutagenesis from this process in later stages of the disease can be detrimental to patient outcomes. This is further corroborated by our observation that SBS17 subclonal increases are linked with a reduction of cytotoxic T cells in the microenvironment, which could lead to more immune evasion.

We also observed that SBS17a/b were elevated in the context of disease progression. As the SBS17b trace can also be the result of the chemotherapy itself when 5-FU is applied and we cannot distinguish mutations occurring before and after therapy in this study, it is possible that part of the signal observed is explained by the preservation of the chemotherapy insult in the surviving cells. However, the same explanation cannot be offered for SBS17a increases, which makes it tantalising to hypothesise that OAC is all the more successful in avoiding therapies due to an enhanced proliferation capacity in the presence of SBS17 mutagenesis (for reasons unknown) before and after chemotherapy. This is a complex question which will need to be addressed in future studies.

APOBEC mutagenesis, the A-T rich motif SBS41 signature and BER impairment appeared most distinctly active after OAC transformation, possibly enabling the activation of progression-specific drivers such as

*PIK3CA* and *KRAS*, and tended to dilute in advanced stages. The higher prevalence of these signatures, in particular of APOBEC, in primary tumours suggests it may inform whether a sample is a tumour rather than Barrett Oesophagus, even though the contribution of the APOBEC process is relatively weak. Further longitudinal studies will be required to investigate whether the presence these signatures can predict the risk of progression in Barrett Oesophagus. Importantly, while mutations arising due to BER deficiency were on average relatively few, they marked a significantly worse patient outcome. Intriguingly, these mutational insults, along with other notably prevalent alterations in NER, HR, TLS and MMR genes, appeared uncoupled from the transcriptional activity of the respective pathways, potentially implicating epigenetic regulation that restores lost function later during cancer evolution which requires further study. Nevertheless, these findings suggest that DDR deficiency phenotypes beyond HR may be an underappreciated prognostic and therapeutic opportunity in a subset of OAC patients. These signatures could be easily ascertained in the clinic in the future through cost-effective methods such as mutREAD[37] or highly sensitive ones like NanoSeq[38].

By scaling up the cohorts of analysed cancer genomes, it is becoming clear that the repertoire of uncovered mutational processes in OAC continues to expand. While the SBS17 process undoubtedly dominates across tumour development stages, and SBS41/DDRD mutagenesis appear particularly important in shaping primary tumours, it is likely that a variety of mutational processes will continue to emerge as acting in a minority of OACs, much like the long tail of cancer drivers. For instance, the SBS93 mutational process appeared in some of our solutions although not the optimally chosen one, and it is likely that it will become more significant in larger cohorts since it is also present in gastric cancer and oesophageal squamous cell carcinoma, with its aetiology still to be resolved. While some of the signatures uncovered may provide some clinical utility in the long-term, including for prognosis or delaying progression e.g., by acting with CDK6 inhibitors to supress proliferation early in the disease,

comprehensive longitudinal studies with matched samples across different disease stages will be crucial to elucidate the entire dynamic complexity of these processes. Despite the relatively large size of the cohort in the present study, the findings should be interpreted taking into account the uncertainty around the contribution of the lesser prevalent signatures, particularly in pre-neoplastic conditions. This is further limited by the fact that we are unlikely to be comprehensively sampling the subclonal heterogeneity of Barrett Oesophagus, which is likely to be rather high, due to the limited sequencing depth. Future studies utilising deep sequencing or clonal lineage tracking will be required to shed further light into the complex pre-neoplastic mutational signature heterogeneity. In addition, our insights into metastatic disease are limited by the small number of metastatic and lymph node samples available for analysis. Experimental validation in vitro and in vivo will be crucial in the future to confirm the mutational signature changes observed at disease boundaries. In addition to the lack of matched samples for most of the cases in our cohort, this study is also limited by the uneven availability of pre- and post-therapy samples (with the latter category dominating). Thus, we are predominantly characterising tumours that are refractory to neoadjuvant chemotherapy for which tissue was still available for sampling, and there is a lesser representation of pre-therapy tumours which responded to therapy. The differences in the biology of these tumours can therefore not be accurately captured and further balanced longitudinal studies are required to dissect these aspects.

Further research is also required to elucidate the role of BER and SBS17 mutagenesis in the progression of OAC, from a genetic and environmental perspective. Within our cohort, we did not find any robust associations between mutational signatures and exposure to risk factors such as alcohol, PPI or NSAID usage, and only a moderate correlation between SBS17 and smoking in primary tumours. Our data suggest a potential weak contribution of smoking to progression to adenocarcinoma, in line with previous epidemiological studies in the field[39,40]. We also find an association between smoking and SBS17-related mutagenesis in non-responders to chemotherapy, but these findings do not imply a causal effect and are highly limited by the lack of a suitably sized longitudinal cohort. Interestingly, we also note there is no strong evidence of the classical smoking signature SBS4 in our cohort, which paints a complex picture of the effects of smoking on the OAC cell of origin. This could be explained by weaker exposure or interaction with other risk factors and repair processes that may differ from the ones encountered in the lung. Longitudinal analyses in larger cohorts will be required to elucidate any definitive links. Finally, the frequent mutational process shifts in tumour subclones should be further investigated in relation to clinical outcomes upon various therapies.

To summarise, we have described multiple processes that shape the evolution of OAC, presenting distinguishable as well as common features from pre-neoplastic to advanced disease (Fig. 8). The lack of major differences in clinical risk factors and signatures from Barrett Oesophagus to OAC might underscore the fact that we are comparing different stages of the same disease process, in keeping with findings from ref. 41. The dynamics observed across disease stages are suggestive of putative shifts in intrinsic and environmental pressures that may influence tumourigenic capacity and the microenvironmental niche. These mutational changes could help inform cancer progression and patient prognosis in a stage-dependent manner.

## Methods
The research performed in this study complies with all relevant ethical regulations. The study was approved by the Cambridge South Research Ethics Committee (REC 07/H0305/52 and 10/H0305/1) and included written individual informed consent. No participant compensation was provided.

### Study cohort
A cohort was assembled comprising 161 Barrett, 777 OACs and 59 metastatic samples that had been collected through a multicentre UK wide study called OCCAMS (Oesophageal Cancer Classification And Molecular Stratification) and undergone whole genome sequencing (WGS) as part of the ICGC-International Cancer Genome Consortium. These included 47 pairs of matched Barrett Oesophagus and primary tumours from the same individuals, and four trios of matched Barrett Oesophagus, OAC and metastases. Part of the OAC tumours (214/777) were collected from the Mutographs study with available clinical annotations.

The assembled cohort comprises 85 female and 560 male patients with OAC, and 26 female and 121 male patients with Barrett Oesophagus, based on self-report. All results presented come from amalgamating human data from both sexes. Sex and gender have not been considered in the design of this study, because OAC has a high male dominance and thus any study looking at differences between male and female cancers would likely be underpowered given the available data. No filtering of the human data was performed based on sex or gender, but we do report this information in Supplementary Table 1 and account for this variable when modelling clinical outcomes. Patient age did not differ significantly between Barrett Oesophagus and OAC cases (median of 67 versus 68, see Supplementary Table 1).

A sample from the Barrett/tumour/metastatic sample and a matched germline reference, which was ideally matched blood or if not available normal squamous oesophagus as far away from the tumour as possible (at least 5 cm), was collected during surgical resection or by an endoscopic biopsy. All samples were snap-frozen.

A systematic pathological review was performed to check the cellularity of the tumour samples using hematoxylin-and eosin-stained sections and only samples with ≥70% cellularity were included. DNA was extracted from frozen tumours using the Allprep DNA/RNA mini kit (Qiagen, Hilden Germany) and DNA from blood was isolated using QIAmp DNA blood maxi kit (Qiagen, Hilden Germany).

### Whole genome sequencing and mutation calling
Paired-end whole genome sequencing at 50X depth for tumours and 30X for matched normal (blood) was performed under contract by Illumina (San Diego,US) as part of the International Cancer Genome Consortium. Quality checks were performed using FastQC (http://www.bioinformatics.babraham.ac.uk/projects/fastqc) and in-house tools.

For mutation calling, sequencing reads were aligned against the reference genome (hg19/Ensembl GRCh37) using the latest version of Burrows-Wheeler alignment algorithm, BWA-MEM. Aligned reads were then sorted into genome coordinate order and by using Picard (http://broadinstitute.github.io/picard) duplicate reads were removed. The Strelka 2.0.15 software[42] was used for calling single nucleotide variants and indels (Supplementary Table 7). Functional annotation of the resulting variants was performed using Variant Effect Predictor.

To validate Strelka calls, we also called ran MuTect2 on 10 randomly selected samples. Somatic variants were called using MuTect2 v4.1.7.0 in matched normal mode with a panel of normals and a population germline resource. Orientation bias priors were obtained using LearnReadOrientationModel before running FilterMutectCalls. Default setting were used throughout.

We obtained very good associations in mutational signature prevalence estimates between Strelka and MuTect2, with lesser certainty only for signatures with <5% prevalence (Supplementary Fig. 30). To account for this uncertainty, we set all mutational signature contributions of <5% to 0 in downstream analyses where a cut-off for prevalence was important (e.g., for survival analysis). The motivation for this is that there is more uncertainty that their contributions will be correctly estimated below 5%, and it is less likely such a contribution would play a major role in shaping the dynamics of OAC development.

When examining cancer drivers, only nonsynonymous SNVs and indels were considered.

Sample purity and ploidy values were estimated from WGS profiles using ascatNgs v2.1[43]. Copy number alterations after correction for estimated normal-cell contamination were also inferred with ascatNgs, using read counts at germline heterozygous positions estimated by GATK 3.2-2[44].

## Mutational signature discovery

Mutational signature discovery in the cohort was performed using SigProfilerExtractor[8]. The optimal signature configuration in the cohort was selected from a range of signature combinations from 5 to 17 based on the highest stability and lowest Frobenius reconstruction error for a signature combination. A total of 14 signatures were identified as the optimal configuration, and this was confirmed by independent analysis using the Bayesian methodology from Sigminer[45]. Once the main mutational processes in the cohort were defined, we used deconstructSigs[19] to infer the mutational contributions of these processes to each sample. Indel signatures were inferred using deconstructSigs on COSMIC signature references.

## Transcription/Replication strand bias

The MutationalPatterns package was used to map the SNVs to either the transcribed or untranscribed strand. Likewise for replication bias, SNVs were assigned to lagging or leading strands[46].

## Nucleosome periodicity

Nucleosome periodicity was analysed as described in Pich et al. Cell 2018. Briefly, nucleosome-centred positions in the human genome were stacked and extended (72 bp each side in zoom in analyses, 1000 in zoom out), and the distribution of mutations across different cohorts was plotted. An expected distribution of mutations was obtained by randomising the observed mutations following the pentamer context. Finally, the relative increase of the mutation rate ((observed-expected)/expected) was calculated across all stacked positions. The relative increase signal-to-noise ratio (SNR) was derived from a discrete time fourier transform derived periodogram and compared across 1000 randomisations obtaining an empirical $p$-value.

## Mutation clonality and timing analysis

To infer subclonality of mutations and mutational processes, we first assessed the likelihood for any sample containing subclonality using the Hartigan's dip test on the distribution of purity-corrected variant allele frequencies. Samples with no significant evidence of deviation from unimodal distribution were deemed as fully clonal. The rest of the samples (51%) were assumed to contain subclonality.

Next, we used MutationTimer[17] to infer the timing (early/late) of every mutation called in each genome as follows: for samples that were assumed to be fully clonal, we ran MutationTimer with default parameters (minimal read support = 3, 0 dispersion) and 100 bootstrap iterations; for samples with evidence of subclonality, we ran MutationTimer with modified input specifying the expected subclonal proportions (calculated from a Gaussian mixture model with two components) and inferred both the clonality and timing of mutations. In both cases, the analysis was performed in a whole-genome doubling conscious manner.

We used the MutationTimer results to split the mutations into clonal/subclonal and early/late and performed mutational signature inference using deconstructSigs again on these separate populations. This allowed us to infer a time and clonality-depedent mutational prevalence of various signatures.

Finally, we corroborated our clonal composition results using TrackSig[47], which identifies cancer cell fractions where mutational signature proportions change. The cases where we observed at least one mutational signature change were in agreement with cases where we observed subclonality using the approaches described above.

## DDR genomic event characterisation

To uncover evidence of DDR impairment in the cohort, we examined nonsynonymous mutations, indels, amplifications, deletions and loss of heterozygosity accumulated in >400 genes across 13 DDR pathways as described in Supplementary Table S3 from ref. 34. Non-synonymous mutations and indel categories included missense, non-sense, stop gained/lost, frameshift/in-frame insertions/deletions, initiator codon variants, incomplete terminal codon variants, 5'/3' UTR variants and transcription factor binding site variants. Amplifications were defined as regions with an average copy number that is double or higher than the average ploidy of the sample (as inferred by ascatNgs). Deletions were identified in regions with a copy number that is half or less than the average ploidy of the sample. Loss of heterozygosity was defined for genes with a complete loss of one copy.

## Positive selection

Groups were defined based on disease stage (Barrett Oesophagus, primary tumour, metastasis) or mutational signature dominance. In the latter case, samples where SBS17a + SBS17b contributed the majority of mutations in a sample were classed as 'SBS17 dominant'; the rest of the samples were categorised as 'Other dominance'. Similarly, samples with evidence of dominant SBS3 + SBS8 exposure were classed as 'HR dominant', while the ones without were grouped separately. The dNdScv tool[48] was run separately on samples from the individual groups in order to infer genes that were under positive selection in the respective group. Finally, genes under positive selection were compared between the groups with/without dominance of a particular mutational signature, and common as well as specifically selected genes were extracted. Among these, cancer driver genes were identified by cross-referencing against the COSMIC Cancer Gene Census database. For genes which had not previously been documented as cancer drivers, we used the GTEx database to confirm their expression in oesophageal/gastric tissue. Olfactory receptors were discarded from the analysis as they are believed to be spurious hits.

## Machine learning for OAC stage classification

We used a gradient boost classifier as implemented by the xgboost package in R to train two models to distinguish Barrett Oesophagus cases from primary tumours, and primaries from metastases, respectively, based on prevalence of all mutational signatures and including clonality and timing as covariates in the model. We split the cohort into 70% for discovery and 30% for validation, and used 5-fold cross-validation in 100 iterations to determine the optimal parameters for the training. The features ranked by importance were visualised using a Shapley plot. The modelling procedure was repeated in a similar manner but with prevalence of signatures detailed based on clonality and timing. The accuracies for testing were 87% and 94%, respectively. The analysis employed the code developed at the following GitHub repository: https://github.com/pablo14/shap-values/blob/master/shap.R. Additionally, we used a random forest classifier as implemented in the randomForest R package to confirm the signature ranking and overall prediction performance.

We also built a multinomial regression model which took as features mutational signature exposures, timing and clonality of signatures and trained a classifier to predict the stage of the tumour (with the 3 stages, Barrett, primaries, metastases, predicted simultaneously). This analysis was implemented using the glmnet package in R.

## RNA sequencing

RNA was quantified using the Qubit High Sensitivity RNA kit (Thermo Fisher) and checked for quality (RNA integrity number; RIN) on the Agilent 2100 Bioanalyzer® (Agilent Technologies, USA) using the RNA

6000 Nano kit. Samples with insufficient material, or an incalculable RIN were excluded. There was no other lower limit for RIN inclusion.

Libraries were prepared with an input of 250 ng RNA using the TruSeq Stranded Total RNA High Sensitivity protocol with ribosomal depletion. Samples with less than the specified input, but with >100 ng total were included and this was noted for the analysis. Library quality and quantity were checked using the Agilent 2100 Bioanalyzer with the DNA 1000 kit and KAPA quantification (KAPA Biosystems, Roche, Switzerland) and were pooled according to the Illumina protocol. Samples were run on the HiSeq 4000 instrument to generate 75 bp paired-end reads. A mixture of normal expression controls was run on each plate: squamous oesophagus, gastric cardia, duodenum. Duodenum mimics the intestinal appearance of Barrett Oesophagus and it is hypothesised that Barrett Oesophagus arises from gastric cells. Squamous oesophagus is a less useful comparison because it shares few features with the glandular epithelium of Barrett Oesophagus.

RNA sequencing data was trimmed for poor quality bases using Trim Galore (https://www.bioinformatics.babraham.ac.uk/projects/trim_galore/) and was then aligned using STAR using the ENSEMBL gene annotation. Reads per gene were quantified using the summariseOverlaps function from the GenomicRanges package, which was also later used for computing Transcripts per million (TPM).

### Hallmarks of cancer and tumour microenvironment signatures

The cancer hallmark signatures were obtained from the CancerSEA database[49]. The tumour microenvironment signatures and composition were inferred using ConsensusTME[50].

Chromosomal instability was calculated as the number of segments with an abnormal copy number (gain/loss) spanning >5% of the length of a chromosome. These numbers were subsequently scaled via a Z-score transformation.

The proliferative capacity of tumours was calculated from RNA-seq data based on markers of G0 arrest as 1-QS, where QS is the combined Z-score of G0 arrest markers as described in Supplementary Table 1 from ref. 32.

### Statistics

Group comparisons were performed using the Student's $t$ test (two-tailed), Wilcoxon rank-sum test or ANOVA, as appropriate. Multiple testing correction using the Benjamini-Hochberg method was performed where appropriate.

Survival analysis was performed using univariate or multivariate Cox Proportional Hazards models as implemented in the ggforest R package. The optimal prognostic cut-offs for mutational signatures were determined using the maximally selected rank statistic, as implemented in the survminer package in R. Kaplan–Meier curves were plotted using the survminer package.

### Reporting summary

Further information on research design is available in the Nature Portfolio Reporting Summary linked to this article.

### Data availability

The raw DNA sequencing data used in this study have all been previously published and are deposited at the European Genome-Phenome Archive (EGA) under accession codes: EGAD00001007785 (whole-genome sequencing of primary tumours and matched normal), EGAD00001006083 (whole-genome sequencing of primary tumours and matched normals), EGAD00001005434 (whole-genome sequencing of primary tumours, Barrett Oesophagus, metastases and matched normals), EGAD00001006349 (whole-genome sequencing of Barrett Oesophagus samples and matched normals). The raw sequencing data are available under restricted access due to data privacy laws; access can be requested to the ICGC Data Access Compliance Office as described here: https://docs.icgc-argo.org/docs/data-access/daco/applying. The processed mutation data for 409 primary tumours employed in this study are also available at the ICGC Data Portal (https://dcc.icgc.org/), under accession code ESAD-UK. The GRCh38/hg38 patch release 13 of the human reference genome [https://www.ncbi.nlm.nih.gov/assembly/GCF_000001405.39/] has been employed in this study. Source data are provided with this paper.

### Code availability

The scripts developed during the analysis presented here are available at the following GitHub repository, released under a GNU GPL-v3.0 license: https://github.com/secrierlab/Mutational-Signatures-OAC (Zenodo https://doi.org/10.5281/zenodo.8063940 [51]). This includes scripts for mutational signature and clonality inference, positive selection analysis, genomic associations, development and testing of mutational signature-based classifiers, DDR pathway analyses and clinical associations.

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

## Acknowledgements

The laboratory of R.C.F. was funded by a Core Programme Grant from the Medical Research Council (RG84369). OCCAMS2 was funded by a Programme Grant from Cancer Research UK (RG81771/84119, A22720/A22131). M.S. was supported by a UKRI Future Leaders Fellowship (MR/T042184/1). S.A. was funded by Cambridge Trust, Trinity College-Henry Barlow Trust and Basil Howard Research studentship from Sidney Sussex College Cambridge. N.L.-B. acknowledges funding from the European Research Council (consolidator grant 682398). IRB Barcelona is a recipient of a Severo Ochoa Centre of Excellence Award from Spanish Ministry of Science, Innovation and Universities (MICINN, Government of Spain) and is supported by CERCA (Generalitat de Catalunya). S.A.Z. is funded by the Gates Cambridge Trust, United Kingdom, and the Jack Kent Cooke Foundation, United States. This research was supported by the NIHR Cambridge Biomedical Research Centre (BRC-1215-20014). The views expressed are those of the authors and not necessarily those of the NIHR or the Department of Health and Social Care.

## Author contributions

M.S. and R.C.F. designed the study and supervised the analyses. S.A. and M.S. conducted the analyses. O.P. provided support on the signature extraction and did the nucleosome periodicity analysis. N.L.-B. supervised the nucleosome periodicity analysis. G.D. constructed and managed the sequencing alignment and variant-calling pipelines. A.K.-S. curated the Barrett Oesophagus cohort, extracted and organised the sequencing of these samples. S.A.Z., S.K., A.K.-S., C.C., B.N. and N.G. provided clinical demographic data. S.A., M.S. and R.C.F. wrote the manuscript, with contributions from all other authors. All authors read and approved the manuscript.

## Competing interests

The authors declare no competing interests.

## Additional information

## OCCAMS Consortium

Rebecca C. Fitzgerald [1,37] ✉, Paul A. W. Edwards[1,3], Nicola Grehan[1], Barbara Nutzinger[1], Elwira Fidziukiewicz[1], Aisling M. Redmond[1], Sujath Abbas[1], Adam Freeman[1], Elizabeth C. Smyth[8], Maria O'Donovan[1,9], Ahmad Miremadi[1,9], Shalini Malhotra[1,9], Monika Tripathi[1,9], Calvin Cheah[1], Hannah Coles[1], Conor Flint[1], Matthew Eldridge[3], Maria Secrier [7,37] ✉, Ginny Devonshire [3], Sriganesh Jammula[3], Jim Davies[10], Charles Crichton[10], Nick Carroll[8], Richard H. Hardwick[8], Peter Safranek[8], Andrew Hindmarsh[8], Vijayendran Sujendran[8], Stephen J. Hayes[11,12], Yeng Ang[11,13,14], Andrew Sharrocks[14], Shaun R. Preston[15], Izhar Bagwan[15], Vicki Save[16], Richard J. E. Skipworth[16], Ted R. Hupp[17], J. Robert O'Neill[8,16,17], Olga Tucker[18,19], Andrew Beggs[18,20], Philippe Taniere[18], Sonia Puig[18], Gianmarco Contino[18], Timothy J. Underwood[21,22], Robert C. Walker[21,22], Ben L. Grace[21], Jesper Lagergren[23,24], James Gossage[23,25], Andrew Davies[23,25], Fuju Chang[23,25], Ula Mahadeva[23], Vicky Goh[25], Francesca D. Ciccarelli[25], Grant Sanders[26], Richard Berrisford[26], David Chan[26], Ed Cheong[27], Bhaskar Kumar[27], L. Sreedharan[27], Simon L. Parsons[28], Irshad Soomro[28], Philip Kaye[28], John Saunders[11,28], Laurence Lovat[29], Rehan Haidry[29], Michael Scott[30], Sharmila Sothi[31], Suzy Lishman[3], George B. Hanna[32], Christopher J. Peters[32], Krishna Moorthy[32], Anna Grabowska[33], Richard Turkington[34], Damian McManus[34], Helen Coleman[34], Russell D. Petty[35] & Freddie Bartlett[36]

[8]Cambridge University Hospitals NHS Foundation Trust, Cambridge CB2 0QQ, UK. [9]Department of Histopathology, Addenbrooke's Hospital, Cambridge, UK. [10]Department of Computer Science, University of Oxford, Oxford OX1 3QD, UK. [11]Salford Royal NHS Foundation Trust, Salford M6 8HD, UK. [12]Faculty of Medical and Human Sciences, University of Manchester, Manchester M13 9PL, UK. [13]Wigan and Leigh NHS Foundation Trust, Wigan, Manchester WN1 2NN, UK. [14]Faculty of Biology, Medicine and Health, University of Manchester, Manchester M13 9PL, UK. [15]Royal Surrey County Hospital NHS Foundation Trust, Guildford GU2 7XX, UK. [16]Edinburgh Royal Infirmary, Edinburgh EH16 4SA, UK. [17]Edinburgh University, Edinburgh EH8 9YL, UK. [18]University Hospitals Birmingham NHS Foundation Trust, Birmingham B15 2GW, UK. [19]Heart of England NHS Foundation Trust, Birmingham B9 5SS, UK. [20]Institute of Cancer and Genomic Sciences, University of Birmingham, B15 2TT Birmingham, UK. [21]University Hospital Southampton NHS Foundation Trust, Southampton SO16 6YD, UK. [22]Cancer Sciences Division, University of Southampton, Southampton SO17 1BJ, UK. [23]Guy's and St Thomas' NHS Foundation Trust, London SE1 7EH, UK. [24]Karolinska Institute, Stockholm SE-171 77, Sweden. [25]King's College London, London WC2R 2LS, UK. [26]Plymouth Hospitals NHS Trust, Plymouth PL6 8DH, UK. [27]Norfolk and Norwich University Hospital NHS Foundation Trust, Norwich NR4 7UY, UK. [28]Nottingham University Hospitals NHS Trust, Nottingham NG7 2UH, UK. [29]University College London, London WC1E 6BT, UK. [30]Wythenshawe Hospital, Manchester M23 9LT, UK. [31]University Hospitals Coventry and Warwickshire NHS, Trust, Coventry CV2 2DX, UK. [32]Department of Surgery and Cancer, Imperial College, London W2 1NY, UK. [33]Queen's Medical Centre, University of Nottingham, Nottingham, UK. [34]Centre for Cancer Research and Cell Biology, Queen's University Belfast, Belfast BT7 1NN, Northern Ireland. [35]Tayside Cancer Centre, Ninewells Hospital and Medical School, Dundee DD1 9SY, Scotland. [36]Portsmouth Hospitals NHS Trust, Portsmouth PO6 3LY, UK.

