## [Peer Review File · Nature Communications]

REVIEWER COMMENTS

Reviewer #1 (Remarks to the Author):

The authors have adequately addressed the concerns in the revised version of the manuscript. There are no further comments and we thank the authors for their research and contributions to the field.

Reviewer #4 (Remarks to the Author), replacement ref for Reviewer #3: Expert in DNA damage and mutational signatures

In this study, the authors analyzed the mutational landscape of esophageal carcinoma progression, in a large cohort consisting of Barrett's esophagus, primary carcinoma and metastasis patients. The authors delineate the mutational processes that can contribute to disease progression and highlight the major differences between disease inception and full-blown tumors/metastasis. Notably, the dynamics of SBS17 serve as a key biomarker of disease stage in these samples.

The paper is well-written overall. The authors have done a satisfactory job of addressing key reviewer questions adequately, including more robust mutation calling, signature-clinical outcomes correlations, and signature correlations with DNA damage response pathways.

This reviewer has a few minor suggestions to further improve the manuscript, which are as follows.

Based on your study and the ones prior (e.g. Alexandrov et. al 2016), there's obviously a strong correlation between esophageal carcinoma and smoking. For both Barrett's and OAC, roughly a fourth of subjects are never-smokers. I am curious as to how the overall mutation loads vary between smokers and never-smokers for these classes of patients. Do you see fewer DDR-associated mutations in never-

smokers? Are there significant differences in mutation signatures in response to treatment between smokers v never-smokers? Based on your current mutational data, can you make a case for smoking to worsen Barrett's prognosis and accelerate cancer progression? A line or two to this effect in either Results or Discussion might illuminate interested readers.

Line 395. MRR is perhaps a typo and should instead be MMR.

Line 395. On a similar note, for the uninitiated, define BER, TLS, MMR, NER at first usage.

Line 728- This paragraph references Pearl et. al (2015) for DDR gene mutational annotation. While the referenced paper does contain the methodology for detecting non-synonymous changes the mutated DDR genes, the authors could perhaps elaborate this section by informing readers about how they identified the various classes of mutations (AMP, LOH etc.) and assigned them to the different genes, including any statistical cut-offs.

Reviewer #5 (Remarks to the Author), replacement ref for Reviewer #2: Expert in oesophageal cancer genomics and therapy

In this manuscript, Abbas, Fitzgerald, Secrier, co-authors, and OCCAMS study the evolution of oesophageal adenocarcinoma mutational signatures using a novel cohort of 161 BO, 777 primary tumours, and 59 metastases. They describe the mutational landscape across these three groups and describe mutation signature dynamics across the groups. My major concern is that the makeup of the samples comprising these three groups limit the interpretation of their data as evolution from pre-malignant, malignant, and metastatic disease.

Major comments:

1. Barrett oesophagus is not a homogeneous entity, nor is it always definitively pre-malignant. Barrett with no dysplasia will be different genetically from low- or high- grade dysplasia. High grade dysplasia may be microscopically malignant already. It is unclear how these samples were obtained. Were they from a non-malignant patient population undergoing screening or was it "tumour-adjacent" BO which would again be more likely to be malignant. This may be a reason for such similar mutational profiles between BO and OAC groups. CNV profiling may help confirm BO and OAC separation.

2. The authors state that OAC specimens were taken from resection specimens. They acknowledge that these tumors will have been exposed to neo-adjuvant therapy. Because these are unmatched samples (mostly), there is another major confounding factor. Patients whose tumors responded completely or near-completely to neo-adjuvant therapy will not have tissue available for sampling and therefore not be part of this study. This biases the study towards tumor biology which is resistant to neo-adjuvant therapy.

3. Following 2. Line 184-7 “Signatures SBS28 and SBS35 are scarcely visible in Barrett Oesophagus (two and one samples, respectively, with an exposure >5%) but they are clearly visible in primary tumours suggesting that they are primarily operative at the stage of invasive disease.” SBS35 is due to neoadjuvant therapy not biology of the tumor. SBS28 is unknown but similar to SBS17b which is implicated in 5FU.

4. Where were the metastatic samples taken from? Were they all from the same metastatic site?

5. In Figure 4, clonality was a strong predictor; could this “subclonality” reflect normal cell contamination in BO? Along those lines, pathologically, there was 70% cellularity. Given the availability of tumour/normal pairings, was a tumour cellularity calculated from the WGS data.

6. “The change in tumour volume (T stage) was calculated from pre-treatment staging to post-therapy resection pathology staging.” This is an unusual way to categorize response. Most tumours will be T3 leaving no real room for progressive disease. Usually, tumor regression by some sort of scale is used – Mandard, Ryan, etc.

Minor Comments:

1. Line 395 – MMR not MRR

Point-by-point response to Reviewers' comments

Reviewer #4

In this study, the authors analyzed the mutational landscape of esophageal carcinoma progression, in a large cohort consisting of Barrett's esophagus, primary carcinoma and metastasis patients. The authors delineate the mutational processes that can contribute to disease progression and highlight the major differences between disease inception and full-blown tumors/metastasis. Notably, the dynamics of SBS17 serve as a key biomarker of disease stage in these samples.

The paper is well-written overall. The authors have done a satisfactory job of addressing key reviewer questions adequately, including more robust mutation calling, signature-clinical outcomes correlations, and signature correlations with DNA damage response pathways.

We thank the Reviewer for the positive comments.

This reviewer has a few minor suggestions to further improve the manuscript, which are as follows.

Based on your study and the ones prior (e.g. Alexandrov et. al 2016), there's obviously a strong correlation between esophageal carcinoma and smoking. For both Barrett's and OAC, roughly a fourth of subjects are never-smokers. I am curious as to how the overall mutation loads vary between smokers and never-smokers for these classes of patients.

The link between smoking and Barrett Oesophagus/OAC progression is indeed an interesting one and we thank the reviewer for the suggestion to explore this more depth. We have expanded the Results section to address all the Reviewer's comments related to smoking, and have created a new section entitled "Risk factors and clinical associations" (lines 279-316), with new Supplementary Figures 14-18 where we describe these new results as well as elaborate on associations with other risk factors which we had only briefly mentioned previously. Indeed, never smokers comprise a considerable fraction of our cohort (23% of Barrett Oesophagus and primary tumours), and thus we are reasonably powered to capture significant associations with genomic variables. In fact, we find an increase in SBS17a/b levels in past/present smokers compared to never smokers in primary tumours, although not in Barrett Oesophagus - see new Supplementary Figures 14-15 and lines 295-297:

"Current/past smokers presented increased levels of SBS17a/b compared to never smokers in the primary tumours, but not in Barrett Oesophagus (Supplementary Figure 15)."

In terms of mutational load, we did not see a significant difference between never smokers and past/present smokers, neither in Barrett Oesophagus nor in OAC samples, although there was a slight trend for higher mutational burden in the smokers category (see the new Supplementary Figure 16a). Mutational burden also did not differ when further sub-categorising by Barrett Oesophagus grade (dysplastic/non-dysplastic), as shown in Supplementary Figure 16b. We describe this in the Results (lines 297-298):

"Mutational loads did not vary by smoking status at any pre-malignant or cancer stage (Supplementary Figure 16)."

Do you see fewer DDR-associated mutations in never-smokers?

There are fewer patients with SNVs in genes involved in direct repair after multiple testing correction (Fisher's exact test adjusted $p=0.046$, odds ratio=0.15, see new Supplementary

Figure 17). Before multiple testing correction there were also lower fractions of indels in the TLS pathway, fewer amplifications in the S phase genes, and an enrichment of amplifications in ubiquitins and ubiquitin-like proteins. Nevertheless, these associations disappear after multiple testing correction. Overall, we would conclude there may be some weak signals of a protective effect on certain DDR pathways in never smokers, but no strong evidence. The mutation fractions per category, split by smokers and never smokers are included in Supplementary Figures 17, and we have also updated the Results text accordingly (lines 299-303):

“DNA damage repair associated mutations were also broadly similar (Supplementary Figure 17), with a marginal depletion of SNVs affecting genes involved in direct repair in never smokers (Fisher’s exact test adjusted $p < 0.05$, odds ratio=0.15). Overall, no strong signals of a protective effect from mutagenesis appeared to be present in never smokers in pre-cancerous stages.”

Are there significant differences in mutation signatures in response to treatment between smokers v never-smokers?

Unfortunately we do not have pre/post treatment data that would allow us to capture changes in mutational signatures as a result of treatment. However, we have looked at whether post-treatment signatures were different after therapy between smokers and non-smokers, and we find increased SBS17a in past/present smokers with stable or progressive disease after neoadjuvant chemotherapy compared to never-smokers (see new Figure 7d). No difference was observed for complete/partial responders to the therapy (Supplementary Figures 28a, 29a). A similar trend was observed in the context of radiotherapy, although it did not pass the statistical significance cut-off (Supplementary Figures 28b, 29b). We highlight this in the main manuscript (lines 503-509):

“When examining poor responders to neoadjuvant chemotherapy, we found that past or present smokers showed an increased SBS17a mutagenesis signal in their tumours after treatment compared to never smokers (Figure 7d). No differences were observed in individuals presenting complete or partial response to chemotherapy by smoking status (Supplementary Figures 28-29). A similar trend was observed in tumours in the context of radiotherapy, but these did not reach statistical significance (Supplementary Figures 28b, 29b).”

However, the lack of matched samples hampers this analysis and we have highlighted this in the Discussion (lines 632-634):

“We also find an association between smoking and SBS17-related mutagenesis in non-responders to chemo- and radiotherapy, but these findings do not imply a causal effect and are highly limited by the lack of a suitably sized longitudinal cohort.”

Based on your current mutational data, can you make a case for smoking to worsen Barrett’s prognosis and accelerate cancer progression? A line or two to this effect in either Results or Discussion might illuminate interested readers.

We cannot draw any conclusions with respect to a causal effect of smoking on Barrett’s prognosis, but we do see a significantly lower fraction of smokers in the non-dysplastic Barrett Oesophagus patients that do not go on to progress to cancer compared to all the other categories (Fisher’s exact test $p = 0.009$, odds ratio = 0.24, Supplementary Figure 18), in keeping with other studies in the field. We have updated these results in the text (lines 303-306):

“However, we did observe a significantly lower fraction of smokers in the non-dysplastic Barrett Oesophagus patients that do not go on to progress to cancer compared to all the other categories (Fisher’s exact test $p = 0.009$, odds ratio = 0.24, Supplementary Figure 18). This is in keeping with smoking being a known risk factor for progression to OAC³⁰.”

We have also added a few lines in the Discussion to comment more broadly on the link between smoking and OAC reflecting on the results obtained (lines 627-639 / 649-651):

“Within our cohort, we did not find any robust associations between mutational signatures and exposure to risk factors such as alcohol, PPI or NSAID usage, and only a moderate correlation between SBS17 and smoking in primary tumours. Our data suggest a potential weak contribution of smoking to progression to adenocarcinoma, in line with previous epidemiological studies in the field^{39,40}. We also find an association between smoking and SBS17-related mutagenesis in non-responders to chemo- and radiotherapy, but these findings do not imply a causal effect and are highly limited by the lack of a suitably sized longitudinal cohort. Interestingly, we also note there is no strong evidence of the classical smoking signature SBS4 in our cohort, which paints a complex picture of the effects of smoking on the oesophageal adenocarcinoma cell of origin. This could be explained by weaker exposure or interaction with other risk factors and repair processes that may differ from the ones encountered in the lung. Longitudinal analyses in larger cohorts will be required to elucidate any definitive links.

[.] The lack of major differences in clinical risk factors and signatures from Barrett Oesophagus to OAC might underscore the fact that we are comparing different stages of the same disease process, in keeping with findings from Nowicki-Osuch et al⁴¹.”

Line 395. MRR is perhaps a typo and should instead be MMR.

We thank the Reviewer for highlighting this typo. We have now corrected it.

Line 395. On a similar note, for the uninitiated, define BER, TLS, MMR, NER at first usage.

We apologise for this oversight. While some of the pathways (NER, MMR) had already been spelled out at first use earlier in the text, not all of the pathways mentioned in line 395 had been explained before. We have therefore defined all of them in this sentence for ease of comprehension (now in lines 425-427):

“Among the most frequently altered pathways were base excision repair (BER), nucleotide excision repair (NER), HR, translesion synthesis (TLS), Fanconi Anemia, mismatch repair (MMR) and non-homologous end joining (NHEJ)...”

Line 728- This paragraph references Pearl et. al (2015) for DDR gene mutational annotation. While the referenced paper does contain the methodology for detecting non-synonymous changes the mutated DDR genes, the authors could perhaps elaborate this section by informing readers about how they identified the various classes of mutations (AMP, LOH etc.) and assigned them to the different genes, including any statistical cut-offs.

We thank the Reviewer for this suggestion, which is indeed crucial to ensure the methodology is clear and complete. We have now added the suggested details to the paragraph (lines 767-776):

“To uncover evidence of DDR impairment in the cohort, we examined nonsynonymous mutations, indels, amplifications, deletions and loss of heterozygosity accumulated in >400 genes across 13 DDR pathways as described in Pearl et al³⁴. Non-synonymous mutations and indel categories included missense, non-sense, stop gained/lost, frameshift/in-frame insertions/deletions, initiator codon variants, incomplete terminal codon variants, 5'/3' UTR variants and transcription factor binding site variants. Amplifications were defined as regions with an average copy number that is double or higher than the average ploidy of the sample (as inferred by ascatNgs). Deletions were identified in regions with a copy number that is half or less than the average ploidy of the sample. Loss of heterozygosity was defined for genes with a complete loss of one copy.”

Reviewer #5

In this manuscript, Abbas, Fitzgerald, Secrier, co-authors, and OCCAMS study the evolution of oesophageal adenocarcinoma mutational signatures using a novel cohort of 161 BO, 777 primary tumours, and 59 metastases. They describe the mutational landscape across these three groups and describe mutation signature dynamics across the groups. My major concern is that the makeup of the samples comprising these three groups limit the interpretation of their data as evolution from pre-malignant, malignant, and metastatic disease.

The Reviewer makes a very good point that the make-up of the cohort limits our ability to infer mutation-driver evolution of this disease in a completely unbiased manner, and this is something that should be further emphasised in the manuscript. We have now made changes according to the Reviewer's suggestions below to discuss this aspect more thoroughly and highlight the limitations of this study.

Major comments:

1. Barrett oesophagus is not a homogeneous entity, nor is it always definitively pre-malignant. Barrett with no dysplasia will be different genetically from low- or high- grade dysplasia. High grade dysplasia may be microscopically malignant already. It is unclear how these samples were obtained. Were they from a non-malignant patient population undergoing screening or was it "tumour-adjacent" BO which would again be more likely to be malignant. This may be a reason for such similar mutational profiles between BO and OAC groups. CNV profiling may help confirm BO and OAC separation.

We agree with the Reviewer that the heterogeneity of BO could render the interpretation of the results we obtained more difficult. We have now added more information on the split between non-dysplastic (progressor/non-progressor), low grade, high grade, intramucosal carcinoma and BO adjacent to tumour in Supplementary Table 1. We have also checked whether individual mutational signature contributions varied significantly based on the type of BO sampled, and we see no significant differences, with similar ranges for all categories (see the new Supplementary Figure 6a). Furthermore, following the Reviewer's advice on using CNV profiling to confirm BO and OAC separation, we indeed observe that copy number profiles are similar across all BO categories with a marked shift when moving to primary disease (see new Supplementary Figures 6b-c) – suggesting that the BO samples employed in this study are genomically similar and thus the BO heterogeneity is unlikely to bias the results of the mutational signature analysis. We have updated the Results to include these new data (lines 185-194):

“Despite the fact that the Barrett samples encompassed an entire spectrum from non-dysplastic non-progressors and pre-progressors to low/high grade dysplasia, intramucosal carcinoma and Barrett Oesophagus adjacent to the cancer, the signature prevalence did not differ significantly across these categories (Supplementary Figure 6a), which were also clearly genomically distinct from the primary cancer, with copy number and ploidy profiles in line with those expected in pre-malignant disease (Supplementary Figure 6b-c). Thus, the presence of most signatures early in Barrett Oesophagus samples is unlikely to be due to a confounding effect of malignancy already existing in some of the more advanced cases given the heterogeneity of this cohort, but rather due to most of these processes acting very early on before tumour establishment.”

2. The authors state that OAC specimens were taken from resection specimens. They acknowledge that these tumors will have been exposed to neo-adjuvant therapy. Because these are unmatched samples (mostly), there is another major confounding factor. Patients whose tumors responded completely or near-completely to neo-adjuvant therapy will not have tissue available for sampling and therefore not be part of this study. This biases the study towards tumor biology which is resistant to neo-adjuvant therapy.

The lack of matched samples for most cases in the cohort is a limitation of our study which we have already acknowledged in the manuscript (lines 214-218):

“Nevertheless, it is worth noting that most of the primary tumour and metastatic samples analysed were not originating from the same patients, and for the cases where matched samples were available the increasing/decreasing trends were less clear. Larger cohorts of matched primary-metastatic cases will be needed in the future to further investigate these patterns.”

The Reviewer is right in pointing out the additional bias towards patients who do not respond to neoadjuvant therapy. While we do include samples taken before therapy, the majority of cases are sampled after therapy and this indeed makes it difficult to capture accurately the biology of responders to such therapies. While understanding the differences between responders and non-responders was outside the scope of this study, this limitation should be highlighted, and we have now emphasised it in the Discussion (lines 617-624):

“In addition to the lack of matched samples for most of the cases in our cohort, this study is also limited by the uneven availability of pre- and post-therapy samples (with the latter category dominating). Thus, we are predominantly characterising tumours that are refractory to neoadjuvant chemotherapy for which tissue was still available for sampling, and there is a lesser representation of pre-therapy tumours which responded to therapy. The differences in the biology

of these tumours can therefore not be accurately captured and further balanced longitudinal studies are required to dissect these aspects.“

3. Following 2. Line 184-7 “Signatures SBS28 and SBS35 are scarcely visible in Barrett Oesophagus (two and one samples, respectively, with an exposure >5%) but they are clearly visible in primary tumours suggesting that they are primarily operative at the stage of invasive disease.” SBS35 is due to neoadjuvant therapy not biology of the tumor. SBS28 is unknown but similar to SBS17b which is implicated in 5FU.

The Reviewer is right to emphasise the nature of these signatures as likely related to treatment. We have altered the text to make it clear these signatures would not be expected in BO and instead are not surprising to be seen in primary tumours, especially those that might be treated with 5-FU (lines 198-204):

“This is expected given that SBS35 is linked with platinum treatment, while SBS28 has been linked with polymerase epsilon mutations but also shares similarities with SBS17b and thus could also explain a noisier or imperfectly deconvolved signal in the already established malignancy, possibly also due to 5-FU treatment.”

4. Where were the metastatic samples taken from? Were they all from the same metastatic site?

The samples were derived from distinct lymph node and metastasis sites and we have now provided a breakdown of these sites in Supplementary Table 2. The numbers per site are unfortunately too small to empower any meaningful comparisons by site, a limitation which we highlight in the Discussion (lines 614-616):

“In addition, our insights into metastatic disease are limited by the small number of metastatic and lymph node samples available for analysis.”

5. In Figure 4, clonality was a strong predictor; could this “subclonality” reflect normal cell contamination in BO? Along those lines, pathologically, there was 70% cellularity. Given the availability of tumour/normal pairings, was a tumour cellularity calculated from the WGS data.

The Reviewer makes a pertinent point regarding the confounding effects tumour purity might have on the results. We have indeed calculated tumour purity from WGS data using ascatNgs

and employ it when determining the likelihood of each sample being clonal/subclonal, as specified in the Methods (lines 742-744):

“To infer subclonality of mutations and mutational processes, we first assessed the likelihood for any sample containing subclonality using the Hartigan’s dip test on the distribution of purity-corrected variant allele frequencies.”

Purity is also employed internally by MutationTimer when inferring clonal and subclonal changes within the tumour.

We have now clarified the tool employed to estimate sample purity in the Methods (line 703):

“Sample purity and ploidy values were estimated from WGS profiles using ascatNgs v2.1⁴³.”

Furthermore, following the Reviewer’s suggestion we have checked how tumour purity inferred from WGS differs between Barrett Oesophagus and primary tumours in samples that show any evidence of subclonality versus those that do not (see new Supplementary Figure 23). While we see a slight decrease in purity for fully clonal samples, potentially due to higher immune recognition/infiltration of a more homogeneous cancer entity, the range of purity is similar between primary tumours and BO samples in either case, with no significant differences. This suggests that it is not simply a difference in purity between BO and primary tumours that is picked up by our model presented in Figure 4, but rather a genuine effect of the clonal or subclonal action of specific mutational processes. We have updated the text with these new results (lines 385-391):

“Interestingly, it emerged from the model that the clonality of the mutations had a strong contribution to distinguishing between cancer stages (Figure 4b). This was despite the fact that Barrett Oesophagus and primary tumour samples had similar purities both when fully clonal as well as when presenting subclonally (Supplementary Figure 23). Thus, the differences in clonality picked up by the model are unlikely to simply reflect normal cell contamination in Barrett Oesophagus but rather a genuine effect of the clonal or subclonal action of specific mutational processes.”

6. “The change in tumour volume (T stage) was calculated from pre-treatment staging to post-therapy resection pathology staging.” This is an unusual way to categorize response. Most tumours will be T3 leaving no real room for progressive disease. Usually, tumor regression by some sort of scale is used – Mandard, Ryan, etc.

The Reviewer rightfully pointed out that the Mandard classification is the more standard way to look at treatment response. We have now checked this in our analyses, and we do see an increase in SBS17a mutagenesis in tumours with Mandard TRG 3 or higher compared to TRG 1-2 tumours when looking at treatment naive samples only (see new Figure 7b). There are no notable differences when comparing tumours with different TRG scores if we combine the pre- and post-treatment samples together (Supplementary Figure 27). The change in tumour volume calculation is provided as an additional measure, as we could calculate the change in tumour volume for a higher number of cases given the available data thus allowing a more complete assessment of responses via this measurement. We have also updated Figure 7c to illustrate the changes in SBS17b by change in tumour volume when splitting into pre- and post-treatment samples, as this is a fairer way to analyse the data. The trends remain the same as previously reported. As the Reviewer suggested, more cases were observed with tumour shrinkage (93) compared to tumour growth (41) in T3/4 tumours, while in TRG1/2 patients we observed 41 cases of tumour shrinkage versus 52 of growth. Overall, based on multiple measurements (see also Figure 7d for therapeutic response in smokers), it appears that increases in SBS17 mutagenesis tend to be associated with worse clinical outcomes, although this does not imply a causal effect and needs further investigation. We have updated the Results section to clarify these aspects and highlight the limitation in observing progressive disease from T3 onwards (lines 497-514):

“SBS17a and SBS17b exposures did not show significant associations with overall survival outcomes (Supplementary Table 6). However, patients with worse tumour regression outcomes, i.e. Mandard TRG 3 or higher, presented increased SBS17a mutagenesis in the tumour before treatment (Figure 7b, Supplementary Figure 27). We further assessed tumour progression by growth in tumour volume from pre-treatment staging to post-therapy resection, and detected an increased SBS17b prevalence in tumours that grew after surgery (Figure 7c). This was observed both in tumours sequenced before as well as after treatment, potentially hinting at an early SBS17 mutagenic link with patient outcomes. When examining poor responders to neoadjuvant chemotherapy, we found that past or present smokers showed an increased SBS17a mutagenesis signal in their tumours after treatment compared to never smokers (Figure 7d). No differences were observed in individuals presenting complete or partial response to chemotherapy by smoking status (Supplementary Figures 28-29). A similar trend was observed in tumours in the context of radiotherapy, but these did not reach statistical significance (Supplementary Figures 28b, 29b). This, in conjunction with our previous observations that SBS17 signatures tend to be more prevalent in faster proliferating tumours, could indicate a role of this mutational process in conferring more aggressive phenotypes that are also more resistant

to standard therapies for OAC. These observations should nevertheless be considered in light of the dominance of stage T3/4 tumours in our cohort (76% of cases), which limits the chance to observe progressive disease. “

Minor Comments:

1. Line 395 – MMR not MRR

We thank the Reviewer for highlighting this typo. We have now corrected it.

REVIEWERS' COMMENTS

Reviewer #4 (Remarks to the Author):

The authors have adequately addressed all my queries raised in the previous review. The presentation of the data is clearer and the methods are described in satisfactory detail. Overall this is a really nice study.

Reviewer #5 (Remarks to the Author):

Thank you for addressing my comments in such a detailed manner. I think the changes are acceptable and congratulate the authors on performing this large scale study.

Point-by-point response to Reviewers' comments

Reviewer #4 (Remarks to the Author):

The authors have adequately addressed all my queries raised in the previous review. The presentation of the data is clearer and the methods are described in satisfactory detail. Overall this is a really nice study.

Reviewer #5 (Remarks to the Author):

Thank you for addressing my comments in such a detailed manner. I think the changes are acceptable and congratulate the authors on performing this large scale study.

We thank the Reviewers for their very helpful comments which have greatly improved the manuscript.